# Nanoparticle-based targeting of microglia improves the neural regeneration enhancing effects of immunosuppression in the zebrafish retina

Kevin Emmerich [1,2,8], David T. White[2,8], Siva P. Kambhampati[2,3,8], Grace L. Casado[2], Tian-Ming Fu[4,5], Zeeshaan Chunawala[2], Arpan Sahoo [2], Saumya Nimmagadda[2], Nimisha Krishnan[2], Meera T. Saxena[2], Steven L. Walker[6], Eric Betzig [4✉], Rangaramanujam M. Kannan [2,3✉] & Jeff S. Mumm [1,2,3,7✉]

Retinal Müller glia function as injury-induced stem-like cells in zebrafish but not mammals. However, insights gleaned from zebrafish have been applied to stimulate nascent regenerative responses in the mammalian retina. For instance, microglia/macrophages regulate Müller glia stem cell activity in the chick, zebrafish, and mouse. We previously showed that post-injury immunosuppression by the glucocorticoid dexamethasone accelerated retinal regeneration kinetics in zebrafish. Similarly, microglia ablation enhances regenerative outcomes in the mouse retina. Targeted immunomodulation of microglia reactivity may therefore enhance the regenerative potential of Müller glia for therapeutic purposes. Here, we investigated potential mechanisms by which post-injury dexamethasone accelerates retinal regeneration kinetics, and the effects of dendrimer-based targeting of dexamethasone to reactive microglia. Intravital time-lapse imaging revealed that post-injury dexamethasone inhibited microglia reactivity. The dendrimer-conjugated formulation: (1) decreased dexamethasone-associated systemic toxicity, (2) targeted dexamethasone to reactive microglia, and (3) improved the regeneration enhancing effects of immunosuppression by increasing stem/progenitor proliferation rates. Lastly, we show that the gene *rnf2* is required for the enhanced regeneration effect of D-Dex. These data support the use of dendrimer-based targeting of reactive immune cells to reduce toxicity and enhance the regeneration promoting effects of immunosuppressants in the retina.

[1] McKusick-Nathans Institute of the Department of Genetic Medicine, Johns Hopkins University School of Medicine, Baltimore, MD, USA. [2] Department of Ophthalmology, Wilmer Eye Institute, Johns Hopkins University, Baltimore, MD, USA. [3] The Center for Nanomedicine, Wilmer Eye Institute, Johns Hopkins University, Baltimore, MD, USA. [4] Janelia Farms Research Campus, Howard Hughes Medical Institute, Ashburn, VA, USA. [5] Department of Electrical and Computer Engineering and Princeton Bioengineering Initiative, Princeton University, Princeton, NJ, USA. [6] School of Biomedical Sciences, The Chinese University of Hong Kong, Hong Kong, China. [7] Solomon H Snyder Department of Neuroscience, Johns Hopkins University, Baltimore, MD, USA. [8] These authors contributed equally: Kevin Emmerich, David T. White, Siva P. Kambhampati. ✉email: betzige@janelia.hhmi.org; krangar1@jhmi.edu; jmumm3@jhmi.edu

Neurodegenerative diseases are caused by the progressive loss of discrete neuronal cell types. In mammals, this loss is typically permanent as central nervous system (CNS) neurons do not normally regenerate. Therapies for regenerating lost neurons are therefore needed to restore neural function in patients suffering from this class of diseases. In contrast to mammals, fish exhibit a robust capacity for neuronal regeneration[1,2]. In the zebrafish retina, Müller glia (MG) function as injury-induced multipotent neural stem cells capable of rapidly regenerating all retinal neurons[3,4]. Studies in zebrafish have revealed factors that can stimulate limited levels of neural repair in mice, demonstrating that mammalian MG retains the potential to act as retinal stem cells[5–10]. Collectively, these studies indicate that therapies capable of enhancing MG regenerative capacities could provide a means of replacing neurons lost to retinal neurodegenerative disease, thereby enabling recovery of visual function[11].

Research on neuroimmune interactions has largely focused on either developmental roles or the detrimental consequences of chronic neuroinflammation during neurodegeneration[12,13]. Studies focused on immune system roles during retinal regeneration suggest potential differential functions across species. Studies in the chick initially implicated microglia/macrophages as regulators of MG reactivity and proliferation in response to neuronal cell loss[14,15]. Conversely, we found that immunosuppression had a context-dependent effect on rod photoreceptor regeneration kinetics in the larval zebrafish retina. Exposure to dexamethasone (Dex) prior to retinal cell loss inhibited microglia reactivity, decreased MG proliferation rates, and inhibited the regenerative process by ~40%. However, when applied after the onset of cell loss, Dex accelerated rod cell regeneration kinetics by ~30%[16]. Similarly, co-ablation of microglia and photoreceptors inhibited retinal regeneration in zebrafish larvae. In contrast, microglia ablation was shown to enhance the regenerative potential in the mouse retina[17]. In keeping with our results, Silva et al. found that pre-injury treatment with Dex suppressed microglia activation, cytokine production, MG proliferation, and photoreceptor regeneration in adult fish[18]. Similarly, acute inflammatory responses promote regeneration of the optic nerve in zebrafish, but prolonged Dex treatments inhibited repair[19]. Together, these results suggest that context-dependent modulation of immune system reactivity could be used to promote regenerative responses to neuronal loss.

Unfortunately, systemic immunomodulation is known to incur a range of systemic, often deleterious, side effects[20]. Thus, attributing our findings solely to the immunosuppressive activity of Dex is problematic. Nanoparticle-based targeted delivery of immunosuppressants to reactive immune cells could help to clarify effects during neuronal degeneration and regeneration. In addition, cellular targeting could reduce systemic toxicity issues of otherwise powerful drugs, such as glucocorticoids. Furthermore, nanoparticles can facilitate the transport of small molecule cargo across the blood–brain and blood retinal barriers[21]. For example, we have shown that hydroxyl (poly)amidoamine (PAMAM) dendrimer nanoparticles enable the targeting of drug conjugates to reactive microglia/macrophages, as well as other phagocytic cells such as retinal pigment epithelia (RPE) cells, in multiple inflammation-associated disease models[22–28]. Dendrimers are well-defined tree-like molecules that can be synthesized at a precise size, charge, and molecular weight[29–31]. The surface groups are amenable for conjugation to multiple types of cargo per molecule (e.g., drugs, targeting moieties, biologics, and imaging agents). In the context of the eye, dendrimer-based targeting of Dex to activated macrophages led to improved clinically-relevant measures in a rabbit model of autoimmune dacryoadenitis[32], and more recently showed neuroprotective

effects in the retina in a rat model of glaucoma[27]. Additionally, a hydroxyl dendrimer-drug conjugate is in early clinical trials for the treatment of inflammation associated with COVID-19, and for the treatment of neuroinflammation in childhood cerebral adrenoleukodystrophy (clinical trials NCT04458298 and NCT03500627, respectively). These data demonstrate that dendrimer-drug conjugates can quell neuroinflammation in neurodegenerative contexts[22].

Here, we investigated potential mechanisms by which post-ablation Dex enhances regeneration kinetics and evaluated the effects of conjugating Dex to dendrimer nanoparticles with respect to microglia targeting, microglia reactivity, retinal stem/progenitor cell proliferation rates, and retinal regeneration kinetics. In vivo time-lapse imaging using adaptive optics-corrected lattice light-sheet microscopy (AO-LLSM)[33,34] to provide vastly improved spatiotemporal resolution, revealed Dex-induced changes in microglia reactivity dynamics. Further, we show evidence that dendrimer-conjugated Dex (D-Dex), by targeting Dex to reactive immune cells: reduces toxicity, resolves microglial reactivity, and enhances retinal regeneration kinetics relative to free Dex controls. The latter finding was associated with D-Dex increasing retinal stem/progenitor cell proliferation rates. Finally, a small-scale CRISPR/Cas9-based screen revealed that *rnf2* is required for the regeneration-promoting effects of D-Dex. Together, these findings support the development of nanotechnology-targeted immunomodulation strategies for promoting neuronal regeneration.

## Results

**Post-ablation Dex treatment suppresses microglia reactivity to rod cell death.** A previously published nitroreductase (NTR) expressing transgenic line facilitating prodrug-induced rod cell death, *Tg(rho:YFP Eco. NfsB)gmc500*[35] (hereafter NTR-rod fish), was used to investigate the effects of post-ablation Dex treatment on microglia reactivity, RNA expression, MG proliferation, and rod regeneration kinetics. In this line, expression of a YFP-NTR fusion protein in rod cells enables selective cell ablation following a 24 h treatment with the prodrug metronidazole (Mtz). As shown previously in refs. [16,35–37], treating NTR-rod larvae with 10 mM Mtz from 5 to 6 days post-fertilization (dpf) resulted in rod cell loss and subsequent regeneration, confirmed by time series intravital images taken prior to ablation at 5 dpf (pre-Mtz), after ablation (post-Mtz, 7 dpf), and following regeneration (post-Mtz, 9 dpf; Supplementary Fig. 1a). To further confirm rod cell ablation, larvae treated ±Mtz at 5 dpf were collected at 6 dpf and retinal sections stained for TUNEL (terminal deoxynucleotidyl transferase (TdT)-mediated dUTP nick end labeling), a marker of DNA damage and cell death[38]. Results showed a significant increase in the number of TUNEL$^+$ cells in the outer nuclear layer (ONL) of Mtz-treated retinas compared to non-ablated controls ($p \leq 0.0001$, Supplementary Fig. 1b, c).

Our previous characterization of the effects of immunosuppression on retinal regeneration indicated that Dex could either inhibit or promote rod photoreceptor regeneration, depending on the timing of treatment. Pre-treating with Dex, prior to induction of rod cell ablation, inhibited regeneration; however, when Dex was applied a day after rod cell loss had been initiated, rod photoreceptor regeneration kinetics accelerated[16]. Here, to control for the possibility that Dex treatment stimulates rod cell survival rather than promotes regeneration, i.e., acted as a neuroprotectant, we modified the protocol and measured YFP levels at 7 dpf instead of 9 dpf (Supplementary Fig. 2a). At this timepoint, maximal loss of YFP signal is observed in Mtz-only controls (see Supplementary Fig. 1a). This facilitates tests for neuroprotective effects, as per a large-scale screen we recently

performed to identify compounds promoting rod cell survival[36]. The results showed that treatment with Dex (2.5 μM) after induction of rod cell loss, i.e., following a 24 h Mtz exposure, had no effect on rod cell survival (Supplementary Fig. 2b).

In our prior study, in vivo time-lapse imaging showed that pre-treating with Dex inhibited microglia reactivity to rod cell death (measured by total migration distance as well as displacement)[16]. However, how microglia dynamics were affected by post-ablation Dex treatment, the paradigm that enhanced regeneration, remains unknown. To address this, we used a powerful intravital microscopy technique developed by the Betzig lab, AO-LLSM[33,39], to increase the spatial and temporal resolution of microglia/macrophage dynamics during in vivo time-lapse imaging of larval zebrafish retinas. Specifically, 30–90 micron retinal volumes were scanned at time scales ranging from 30 to 180 sec/interval—i.e., as much as 20 times faster than our prior study—to assess the effects of post-ablation Dex treatment on microglia/macrophage behaviors near dying rod cells.

For AO-LLSM imaging, NTR-rod fish were crossed with a transgenic line labeling microglia/macrophages with a complementary fluorescent reporter (tdTomato, Tg(mfap4.1:Tomato-CAAX)xt6). Dual-labeled transgenic larvae were selected and treated ±Mtz at 5 dpf for 24 h, Mtz was then removed and larvae in each group were treated ±2.5 μM Dex. High-resolution time-lapse imaging of interactions between microglia and rod cells was performed for all four treatment groups starting at 6.5 dpf (12 h post-Dex treatment). Labeled cells were 4D-rendered in Imaris to facilitate automated quantification of three established metrics of microglia/macrophage reactivity: migration speed, displacement, and sphericity. Each metric was averaged across multiple microglia per fish and compared across multiple fish per condition (a total of 66 cells across 21 fish analyzed). Microglia in non-ablated control larvae (untreated), treated with Dex only (+Dex), typically exhibited ramified morphologies, actively extending and retracting cellular processes, but remained relatively non-migratory and did not typically displace to the photoreceptor cell layer, consistent with a non-reactive homeostatic state (Fig. 1a, b and Supplementary Movies 1, 2). Microglia in rod cell ablated larvae (+Mtz) became highly motile, rapidly migrating in and among dying rod cells, with the majority exhibiting an ameboid morphology suggestive of a reactive state (Fig. 1c and Supplementary Movie 3). Conversely, microglia in rod-ablated larvae treated with Dex 24 h after induction of rod cell loss (+Mtz, +Dex) appeared less motile and displayed a mix of ramified and ameboid morphologies, the former extending and retracting processes similar to microglia in untreated control fish (Fig. 1d and Supplementary Movie 4). Interestingly, in the presence of Dex, ramified microglia were observed even in the degenerating rod cell layer (Supplementary Movie 4). Imaris-based quantification showed a statistically significant increase in microglia migration speed in +Mtz rod cell ablated larvae compared to either of the non-ablated controls (untreated, 36.6%, $p = 0.0107$; +Dex, 38.9%, $p = 0.0046$) and to +Mtz, +Dex larvae (58.7%, $p = 0.0051$; Fig. 1e). The latter three conditions showed indistinguishable migration speeds, suggesting Dex suppressed microglia migration in the post-ablation paradigm (Fig. 1e). Similarly, a statistically significant decrease (~29%, $p = 0.0239$) in microglia displacement was observed in +Mtz, +Dex larvae compared to Mtz-only fish (Fig. 1f). Interestingly, no quantifiable changes in relative sphericity were detected across all conditions (Fig. 1g). Consistent with our prior results[16], these data demonstrate increases in microglia migration and displacement in response to rod cell loss. In addition, post-ablation Dex treatment appears to return microglia to a relatively less motile, potentially non-reactive, state closely resembling behaviors observed in non-ablated control retinas.

To control for the possibility that Mtz influences microglia/macrophage activity in the absence of rod cell ablation, intravital time-lapse confocal imaging was performed in Mtz-treated larvae expressing only the microglia-labeling transgene. A 12 h time-lapse sequence initiated immediately after Mtz exposure shows that, qualitatively, no appreciable change in microglia reactivity was evident (Supplementary Movie 5). Image stills over 2 h intervals show that the vast majority of microglia remained ramified and did not migrate toward any particular region of the retina (Supplementary Fig. 3a and Supplementary Movie 5). Next, 4 h time-lapse sequences were used to compare microglia cell behaviors between untreated controls (−Mtz) and +Mtz larvae (imaging initiated at 4 h post-Mtz exposure). Quantification of the same reactivity parameters as above revealed no significant differences across the two groups, demonstrating that Mtz alone has no effect on microglia/macrophage reactivity (Supplementary Fig. 3b). These results parallel prior reports where Mtz alone was found to have no impact on phagocytic measures in larval zebrafish[40].

**Dendrimer-conjugated Dex (D-Dex) reduces Dex-associated toxicity.** Long-term therapeutic use of Dex and other glucocorticoids is associated with complications due to multiple adverse side effects and systemic toxicity[41]. We have previously shown that conjugation of Dex to dendrimer nanoparticles (hereafter, D-Dex) is effective in suppressing inflammation in a rabbit model of dry eye, leading to a clinically-relevant phenotypic improvement[32]. Additionally, D-Dex conjugates were able to reduce side effects induced by free Dex, presumably by targeting Dex to reactive microglia under inflammatory conditions but facilitating efficient clearance in the absence of inflammation[23,30,32]. Accordingly, we tested if D-Dex conjugates reduce Dex-associated systemic toxicity in zebrafish. For this, we conjugated a generation 4 (G4-OH) PAMAM dendrimer to Dex. D-Dex conjugates used in this study were >98% pure and had ~8 molecules of Dex conjugated to the surface hydroxyl groups of the PAMAM dendrimers, as reported previously[42]. To test if D-Dex reduced toxicity in zebrafish, larvae were injected in the pericardium (PC) with free Dex, D-Dex, or dendrimer alone at 5 dpf (injections were necessary as soaking fish in D-Dex showed no evidence of bioactivity). A total of 504 larvae were tested across a two-fold dilution series ranging from 2.5 to 400 μM of Dex and D-Dex (Fig. 2a). Mortality rates for each condition were measured at 7 dpf, two days after injections. At lower concentrations, 2.5 to 25 μM, no death was observed in any of the three groups. However, Dex-injected fish had increased mortality rates at higher concentrations, reaching 79% toxicity at 400 μM ($LD_{50}$ ~207 μM, Fig. 2b). In contrast, no toxicity was observed in the dendrimer or D-Dex injected groups in the 50 to 200 μM range and only a ~5% toxicity rate was seen in the 400 μM D-Dex injected group (Fig. 2b). Chi-square tests were performed to compare toxicity of each condition to the dendrimer control per each timepoint, statistically significant increases in toxicity were observed for free Dex at 100, 200, and 400 ($p = 0.0219$, 0.0002, and <0.0001, respectively) while no differences in toxicity were seen for D-Dex at any timepoint. These findings are in agreement with previous studies in which D-Dex attenuated Dex toxicity[42].

To directly assess dendrimer clearance rates in the absence of cell loss, we investigated the pharmacokinetics of fluorescently-tagged dendrimer particles using longitudinal whole-organism in vivo imaging. FITC-labeled dendrimers (D-FITC) were injected into the PC of 5 dpf zebrafish. Subsequently, fish were imaged by intravital confocal microscopy at 1 and 2 days post injection (dpi). By 1 dpi, D-FITC had accumulated in the kidney (Supplementary Fig. 4a). To assess D-FITC clearance, Imaris

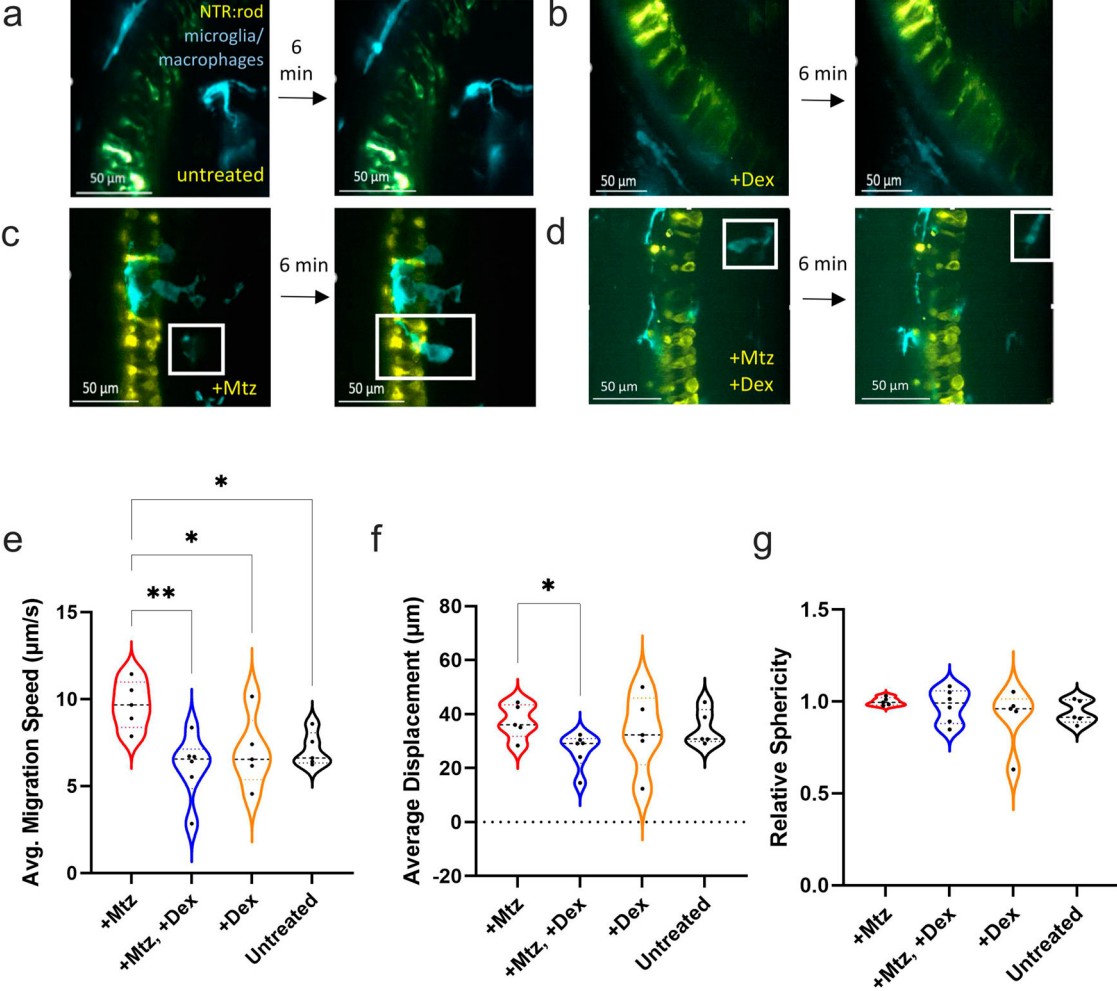

**Fig. 1 Microglia reactivity is altered in response to post-ablation Dex treatment.** **a–d** Image stills 6 min apart from AO-LLSM imaging of 5 or 6 dpf transgenic lines labeling NTR-expressing rods (yellow) and microglia/macrophages (cyan). Four conditions were imaged: non-ablated "untreated" control (A), non-ablated, "+Dex" (B), rod cell ablated, "+Mtz" (C), and rod cells ablated with Dex treatment, "+Mtz, +Dex" (D). In all images, the inner nuclear layer is to the right of the NTR-YFP rod cells. **e–g** Imaris quantification of average migration speed (μm/second), average displacement (μm), and relative sphericity of microglia in each larva, sample sizes for each plot from left to right: 5, 6, 5, 6 with a total of 66 microglia analyzed. See Supplementary Movies 1-4 corresponding to stills A-D, respectively. Asterisks indicate statistically significant differences between the indicated groups (*$p \leq 0.05$, **$p \leq 0.01$), all other comparisons were not statistically significant. Lines within the violin data for each condition for all plots indicate lower quartile (bottom), median (middle line), and upper quartile (top).

was used to quantify D-FITC fluorescent signal in the zebrafish kidney (red circle) at 1 and 2 dpi per each larva. On average, 27% of the D-FITC signal was lost between 1 and 2 dpi demonstrating clearance of injected dendrimers in larval zebrafish in the absence of cell ablation and concomitant microglia reactivity (Supplementary Fig. 4b).

**Dendrimers target reactive microglia following rod cell loss.** Previous static imaging studies have shown that dendrimer nanoparticles accumulate in reactive microglia/macrophages at sites of injury/inflammation[27,32]. The timing and specificity of dendrimer immune cell targeting relative to the onset of injury remain less defined. To better assess the dynamics of dendrimer targeting, in vivo time-lapse confocal imaging of Cy5-conjugated dendrimers (D-Cy5; Fig. 3a)[42] was performed on NTR-YFP larvae treated ±Mtz. To monitor interactions between Cy5-labeled dendrimers, dying rod cells, and immune cells, NTR-YFP fish were combined with another transgenic line expressing tdTomato in macrophage/microglia, *Tg(mpeg1.1:LOX2272-LOXP-tdTomato-LOX2272-Cerulean-LOXP-EYFP)w201*. To promote a

sustained inflammatory environment, cell ablation was prolonged by treating larvae with a decreased concentration of Mtz (2.5 mM) for 48 h, from 5–7 dpf. Midway through this process, at 6 dpf, D-Cy5 was injected pericardially, and time-lapse imaging commenced ~30 min thereafter (Fig. 3b). Time-lapse sequences from +Mtz larvae show frequent evidence of microglia targeting, with D-Cy5 particles observed inside highly migratory cells (Supplementary Movie 6). Zoomed insets from image stills of a representative +Mtz larvae show examples of D-Cy5 particles within retinal microglia (Fig. 3c). Occasionally, in non-ablated control larvae, we also observed non-migratory microglia/macrophages that had engulfed D-Cy5 (Fig. 3d). However, automated IMARIS-based quantification revealed a significant increase in colocalized pixels between D-Cy5 and microglia in Mtz-treated larvae compared to non-ablated controls, indicating rod cell ablation led to increased interactions between dendrimers and microglia ($p = 0.0428$; Fig. 3e). Dendrimer-enhanced targeting of reactive phagocytes thus stands in contrast to nanoparticles, such as silicon dioxide, which show reduced macrophage uptake during inflammation[43]. Finally, we observed instances both in

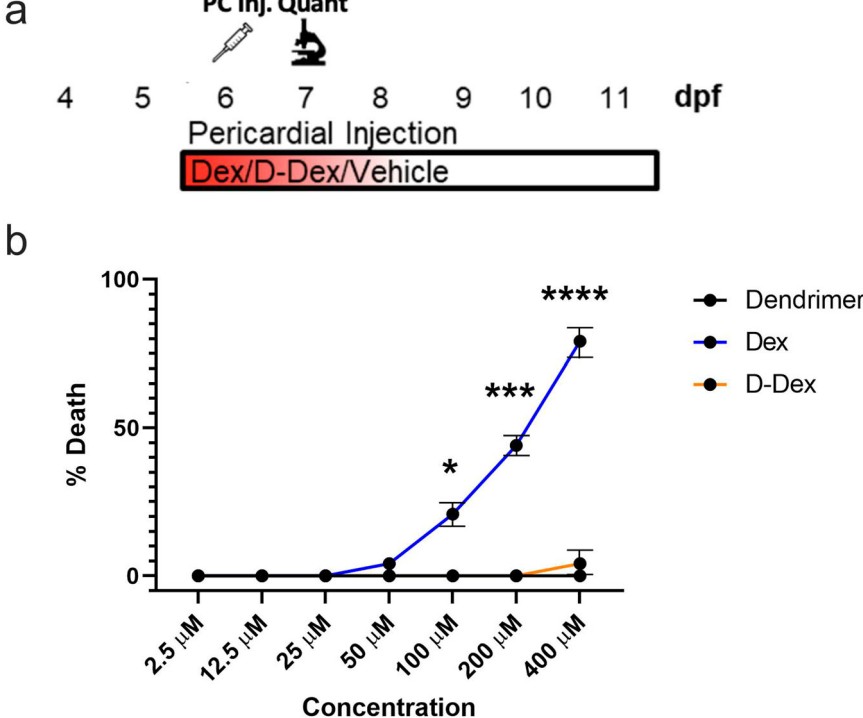

**Fig. 2 Dendrimer conjugation eliminates Dex-mediated toxicity in larval zebrafish. a** Schematic of injection assay to test dendrimer, Dex and D-Dex toxicity. At 6 dpf, dendrimer, "free" Dex, or D-Dex were injected into the pericardium of zebrafish larvae (~10 nL injected volume, 2.5-400 μM for Dex and D-Dex). At 7 dpf, toxicity was quantified based on percent survival across 24 fish per condition over two trials. **b** Line graph indicating average toxicity at 7 dpf for dendrimer (black line), Dex (blue line), and D-Dex (orange line). Comparisons between Dex or D-Dex and dendrimer alone controls showed statistically significant increases in toxicity for Dex-injected larvae only (100-400 μM, *$p \leq 0.05$, ***$p \leq 0.001$, ****$p \leq 0.0001$, respectively).

and out of the retina where D-Cy5 puncta was not colocalized to transgenic microglia/macrophages. These signals suggest D-Cy5 association with other cell types such as dying rods, RPE, MG, or extracellular D-Cy5 that has yet to be taken up. We note that dendrimer uptake by RPE cells has been observed in prior reports[25], suggesting that this may be the source of this signal[25].

**Post-ablation D-Dex treatment improves the regeneration-enhancing effects of immunosuppression.** We previously observed a ~30% increase in rod photoreceptor regeneration kinetics when fish were treated with free Dex 24 h after induction of rod cell ablation[16]. Here, we used the same post-ablation immunosuppression paradigm to compare the effects of free Dex, D-Dex, and unconjugated dendrimers on rod cell regeneration kinetics (Fig. 4a). NTR-rod larvae were exposed to 10 mM Mtz for 24 h from 5–6 dpf to induce rod cell ablation. Mtz was then removed and larvae were either: (1) returned to normal conditions, "+Mtz" controls; (2) immersed in free Dex, "+Mtz, +Dex (soak, 2.5 μM)", control for enhanced regeneration, per our prior study[16]; or injected with either (3) 5 μM Dex, "+Mtz, +Dex (inj)", (4) 5 μM D-Dex, "+Mtz, +D-Dex (inj)", or (5) non-conjugated dendrimers, "+Mtz, +Dendrimer (inj)". At 9 dpf (4 days post-ablation, dpa). YFP+ rod cell regeneration kinetics was measured using an established fluorescent plate reader assay[44]. Compared to Mtz-only controls, all three Dex-treated conditions trended toward enhanced rod cell regeneration, while the dendrimer injected group showed no effect on regeneration kinetics (Fig. 4b). Statistically significant increases were observed for the +Dex (soak) and +D-Dex (inj) conditions. The increase in regeneration kinetics observed with free Dex was ~33% (Fig. 4b, *$p = 0.0459$), in keeping with our previous findings[16]. Intriguingly, larvae injected with D-Dex showed a ~67% enhancement in rod cell regeneration kinetics (Fig. 4b,

****$p = 1.6\text{E-}05$); a statistically significant doubling of the increase in regeneration rate observed with immersion in free Dex (Fig. 4b, **$p = 0.0023$). This result suggests that dendrimer-based targeting of Dex to reactive microglia further enhanced the regeneration-promoting effect of post-ablation Dex treatment.

**Post-ablation D-Dex treatment enhances retinal regeneration kinetics by increasing stem/progenitor cell proliferation.** To explore mechanisms by which D-Dex further enhanced rod cell regeneration, we compared proliferation rates in the retinas of YFP-NTR larvae treated ±Mtz and with or without post-ablation D-Dex injection. Twenty-four hours Mtz exposures from 5 to 6 dpf were followed by pericardial D-Dex injections as above. Larvae were then collected at 8 dpf and processed for DAPI staining and immunolabeled with an antibody against proliferative cell nuclear antigen (PCNA). Larvae treated with Mtz alone showed loss of YFP-expressing rod cells and exhibited mild levels of PCNA labeling (Fig. 4c, upper panel). Non-ablated controls that were injected with D-Dex at 6 dpf showed maintenance of YFP-expressing rod cells and similar levels of PCNA labeling (Fig. 4c, middle panel). Mtz-treated larvae injected with D-Dex exhibited a notable increase in the number of PCNA-expressing cells and evidence of precocious rod cell regeneration (Fig. 4c, lower panel). Quantification of the average number of PCNA+ cells per retina showed a significant increase in the Mtz + D-Dex condition compared to Mtz alone and D-Dex only fish ($p = 2.05\text{E-}02$ and $p = 4.1\text{E-}03$, respectively; Fig. 4d).

Next, to look at the effects of D-Dex on proliferation later in regeneration, larvae were ablated with Mtz and then injected at 6 dpf with either thymine-analog Bromodeoxyuridine (BrdU, 10 mM) alone, or BrdU + D-Dex (5 μM). All larvae were injected with BrdU again at 7 and 8 dpf to ensure robust labeling of proliferating cells. Larvae were collected at 10 dpf for

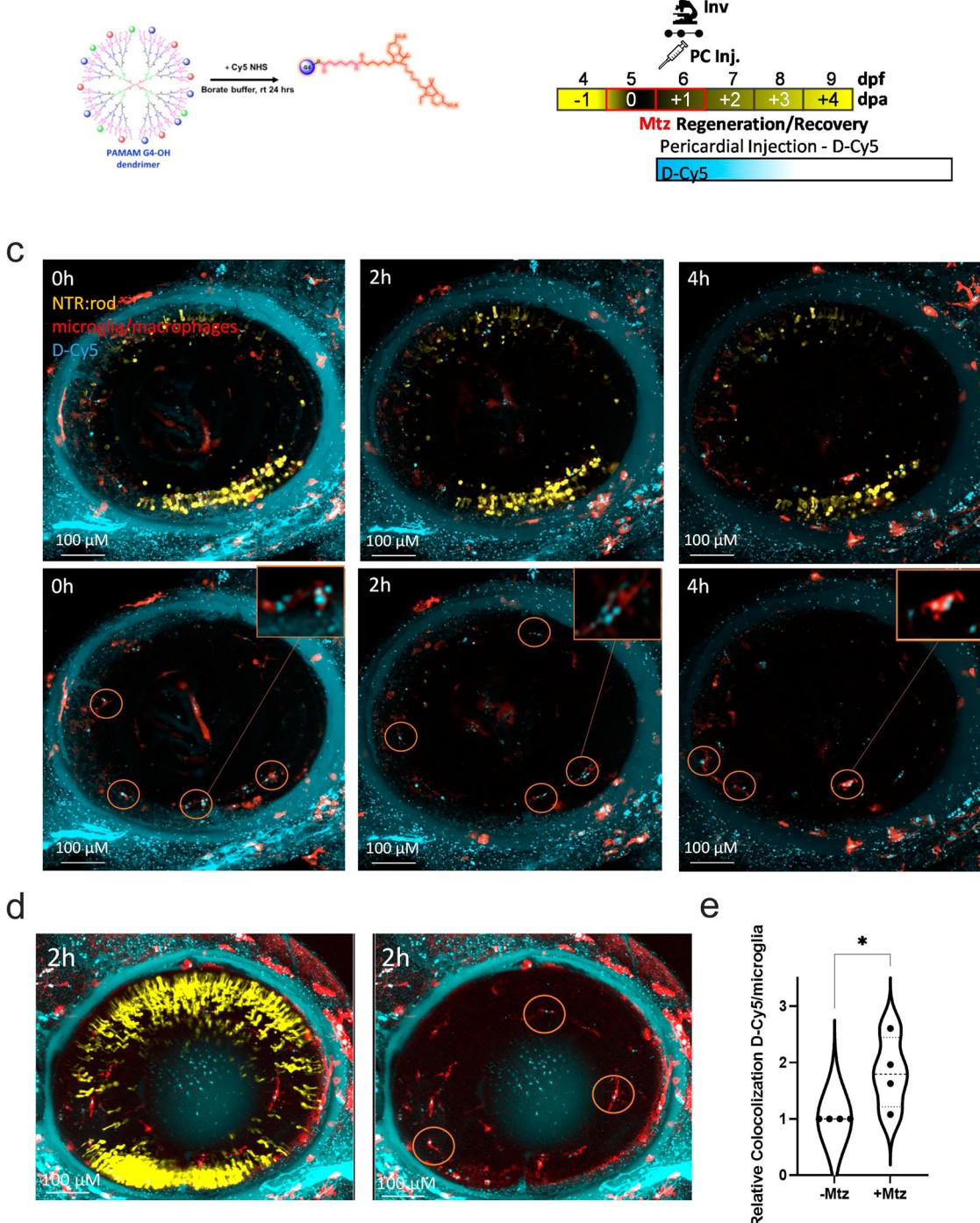

**Fig. 3 Dendrimers localize to reactive microglia following rod cell ablation. a** Process for conjugating Cy5 to PAMAM G4-OH dendrimers (D-Cy5) using Cy5 NHS and borate buffer. **b** Assay schematic: transgenic zebrafish expressing NTR:rod (yellow) and microglia/macrophages (red) were treated with 2.5 mM Mtz from 5–7 dpf to induce rod cell loss. At 6 dpf, fish were given PC injections of D- Cy-5 (cyan, 2 ng/ul) and imaged using in vivo confocal time series microscopy. **c** Image stills from representative larva at 0, 2, and 4 hpi with all three channels (top panels) or with the yellow channel removed (bottom panels); insets and orange circles highlight interactions between dendrimers (cyan) and microglia (red). See Supplementary Movie 6 for the time-lapse sequence. **d** Three (left) and two-channel (right) images still from control non-ablated larva at 2 hpi of D-Cy5; orange circles highlight interactions between dendrimers (cyan) and microglia (red). **e** Imaris-based quantification of normalized colocalization between microglia (red) and D-Cy5 (cyan) signals in retinas treated ±Mtz ($n = 4$ larvae per condition), +Mtz values were normalized to paired sibling (−Mtz) control fish imaged on the same day (*$p \leq 0.05$).

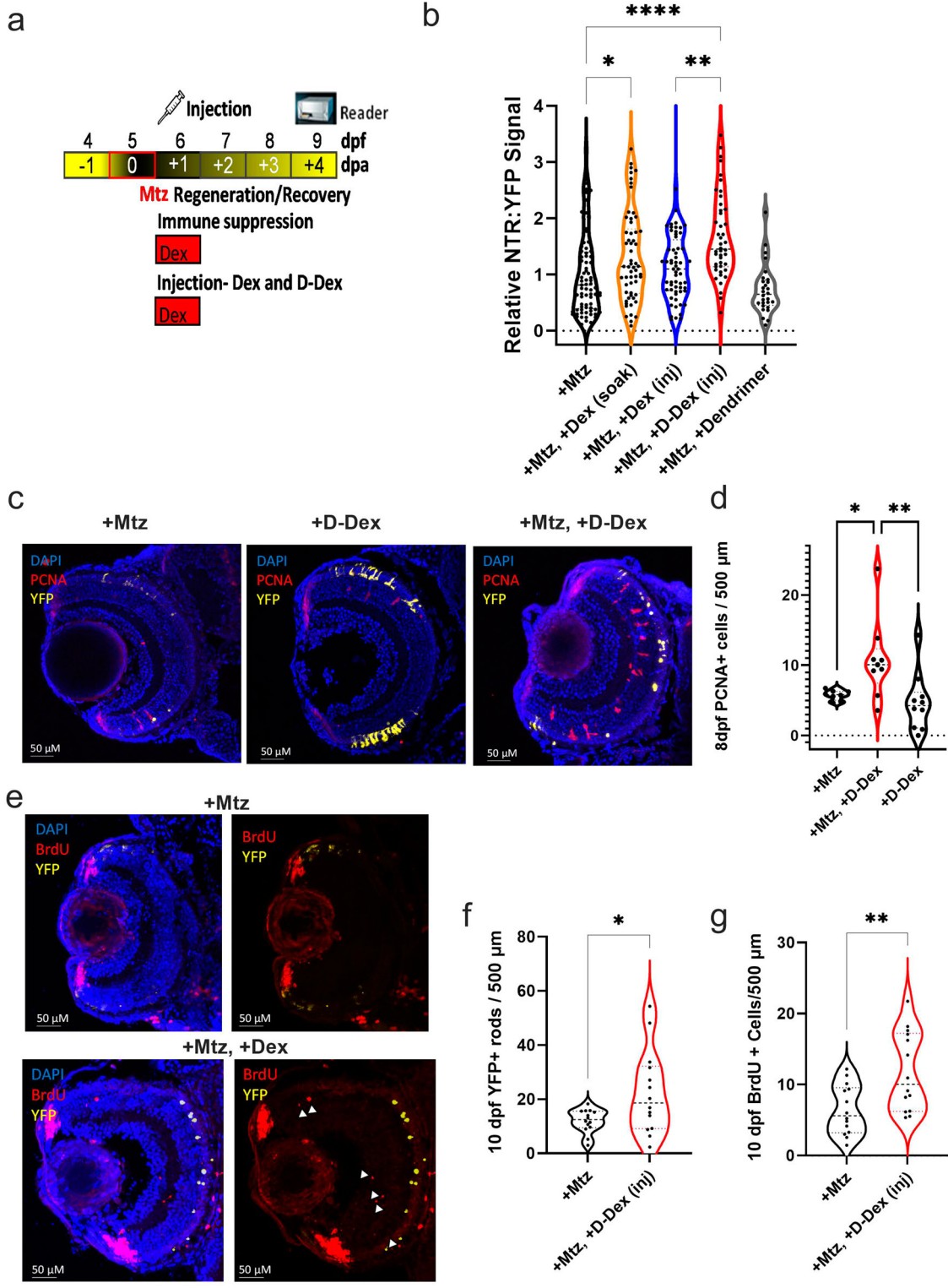

immunostaining with DAPI and anti-BrdU. As expected, for both conditions, prominent BrdU staining was present in the ciliary marginal zone (CMZ), which serves as a control for proliferative marker labeling. However, retinas from the Mtz +D-Dex group exhibited an increase in labeled cells in the inner and outer nuclear layer, where Müller glia (MG) stem-like cells and/or MG-derived progenitors cells reside during retinal regeneration (Fig. 4e). Quantification of the average number of YFP-NTR expressing rod cells showed a significant increase in the Mtz +D-Dex group compared to Mtz alone (Fig. 4f, $p = 2.4E-02$).

Similarly, the number of BrdU-expressing cells in the retina (excluding the lens and CMZ) was also increased in the Mtz +D-Dex group (Fig. 4g, $p = 5.3E-03$).

**Bulk RNA-seq reveals gene expression changes associated with the enhanced regenerative effects of D-Dex.** To investigate the mechanism by which D-Dex further enhanced rod cell regeneration kinetics, we performed bulk RNA-seq to compare: (1) untreated controls, (2) "+Mtz" (10 mM from 5–6 dpf), (3) "+Mtz, +D-Dex" (5 μM at 6 dpf), and (4) "+D-Dex" (5 μM at

**Fig. 4 D-Dex further enhances the regeneration-promoting effects of Dex and induces proliferation. a** Assay schematic for data in (**b**): 5 dpf NTR-rod larvae were exposed to 10 mM Mtz for 24 h and then separated into 5 groups, (1) "+Mtz"; (2) exposed to 2.5 μM free Dex from 6–9 dpf, "+Mtz, +Dex (soak)"; or injected at 6 dpf with either (3) 5 μM free Dex, "+Mtz, +Dex (inj)"; (4) 5 μM D-Dex, "+Mtz, +D-Dex (inj)", or (5) dendrimers alone, "+Mtz, +Dendrimer (inj)". At 9 dpf (4 dpa) NTR-YFP rod signal was quantified by plate reader assay. **b** Quantification of NTR-YFP signals at 9 dpf (4 dpa) to assess rod cell regeneration kinetics, $n = 77, 58, 60, 44,$ and 28 larvae from left to right (*$p \leq 0.05$, **$p \leq 0.01$, ****$p \leq 0.0001$). **c** Representative sections from 8 dpf larvae treated with either 10 mM Mtz only from 5–6 dpf, D-Dex injection only at 6 dpf, or Mtz (5–6 dpf) followed by D-Dex injection at 6 dpf. Images show NTR:rod cells (yellow), DAPI (blue, nuclei), and immunostaining for PCNA (red, proliferating cells). **d** Quantification of PCNA+ cells (not counting the CMZ region), $n = 8, 9,$ and 10 from left to right. **e** Representative sections from 10 dpf larvae treated with either 10 mM Mtz only from 5–6 dpf or Mtz followed by D-Dex injection at 6 dpf. Images show NTR:rod cells (yellow), DAPI (blue, nuclei) and immunostaining for BrdU (red, proliferating cells). **f** Quantification of NTR-YFP-expressing rod cells, $n = 14$ for each group. **g** Quantification of BrdU+ cells, $n = 14$ for each group (*$p \leq 0.05$, **$p \leq 0.01$, ****$p \leq 0.0001$, all others showed non-statistically significant differences).

6 dpf). Eyes were collected at 7 dpf, 24 h after the injection of D-Dex to identify early differentially expressed genes (DEGs) associated with Mtz exposure and/or D-Dex injections (Fig. 5a). Following sequencing, raw data reads were checked for quality and aligned to the zebrafish reference genome. Bioconductor package edgeR was used to identify DEGs between conditions (i.e., genes with a false discovery rate (FDR) of <0.10 and log2 fold change of at least ±1). A subset of top DEGs are plotted across all conditions along with a dendrogram to indicate the consistency of gene expression changes across biological replicates (Fig. 5b).

To identify genes linking D-Dex to enhanced rod regeneration, we focused on the following comparisons: (1) Mtz only vs. untreated, (2) Mtz only vs. +Mtz, +D-Dex, and (3) + D-Dex only vs +Mtz, +D-Dex, yielding 168, 86, and 277 DEGs, respectively (see Supplementary Data 1 for a full list of DEGs). This enabled us to identify DEGs/pathways correlated with D-Dex exposure in the context of both non-ablated and ablated retinas. Functional enrichment analysis between Mtz only and untreated controls revealed upregulated DEGs for Gene Ontology (GO) terms related to apoptosis and acute inflammatory responses and downregulation of DEGs related to the response to light stimulus and eye development (Fig. 5c, top row). Comparisons of +Mtz, + D-Dex to Mtz only retinas revealed upregulated DEGs for GO terms related to stem cell proliferation, retinal development, chromatin-mediated maintenance of transcription, and downregulation of MAPK signaling (Fig. 5c, middle row). Comparisons between +Mtz, +D-Dex, and D-Dex-only controls revealed upregulated DEGs for the GO terms NF-kB signaling and generation of neurons and downregulated DEGs for general immune-related processes. These results are consistent with D-Dex enhancing regeneration kinetics by accelerating the resolution of immune cell reactivity (Fig. 5c, bottom row). We also directly assessed several immune-related factors previously implicated in retinal regeneration[14,45,46] using qRT-PCR to test for D-Dex-associated changes. The results showed upregulation of *il6st* and *stat3*, downregulation of *TNF-alpha*, and no change in *mmp9* expression (Supplementary Fig. 5).

**CRISPR/Cas9 knockdown screen identifies *rnf2* as being required for the regeneration-enhancing effects of D-Dex.** To functionally test a subset of DEGs implicated in D-Dex enhanced rod cell regeneration kinetics, a rapid CRISPR/Cas9-based knockdown screening method[47] was used. Four of the top upregulated DEGs (Mtz+D-Dex vs Mtz alone, average fold change of 8.16) were selected: *rnf2*, *kcnj13*, *col11a2*, and *nosip* (Fig. 6a). Searching an online expression database (ZFIN) for known expression pattern identified known expression in the eye for *rnf2* and *kcnj13*, as well as a known role in immune system function. Less information was available for *col11a2* and *nosip*; furthermore, the knockdown of these two led to developmental abnormalities that precluded further testing. To assess *rnf2* and *kcnj13* function, we first used qRT-PCR to confirm upregulation in ablated larvae injected with D-Dex (Fig. 6b). To test for roles

during rod cell regeneration, F0 "crispant" larvae were created—i.e., larvae exhibiting efficient and widespread somatic knock-down *rnf2* and *kcnj13* expression[47]. Crispant and control larvae underwent rod cell ablation at 5 dpf (10 mM Mtz, 24 h), and were then given a pericardial injection of D-Dex at 6 dpf (5 μM). Rod cell regeneration was assessed at 9 dpf, as above. The effect of *kcnj13* knockdown was minimal, suggesting no specific role in rod cell regeneration (Fig. 6d). In contrast, *rnf2* knockdown larvae showed significantly reduced rod regeneration at 9 dpf compared to +Mtz, +D-Dex controls (Fig. 6c, d, $p = 3.45\text{E-}02$). This data demonstrate *rnf2* is required for D-Dex enabled acceleration of rod cell regeneration kinetics.

## Discussion

Our prior study suggested microglia play a biphasic role during retinal regeneration in zebrafish larvae- initial reactivity is required to stimulate the regenerative process, while subsequent post-injury immunosuppression results in accelerated regeneration kinetics[16]. Recent studies have confirmed that initial inflammatory responses[48] promote retinal regeneration[18,49,50]; however, cellular and molecular mechanisms underlying the pro-regenerative effects of post-injury immunosuppression remain unresolved. Here, combining in vivo time-lapse imaging and immunohistology to investigate cellular responses, we found evidence that post-injury Dex inhibited microglia reactivity and/or accelerated microglia resolution, and increased proliferation of presumptive MG and MG-derived progenitors Conjugation to dendrimer nanoparticles served to target Dex to reactive microglia (and possible other phagocytes such as RPE), reduced systemic toxicity and improved the pro-regenerative effects of post-injury immunosuppression. Combined with recent findings that: (1) macrophages/microglia have been implicated in multiple regenerative paradigms in the zebrafish retina[45,48,49,51,52], (2) selective rod cell ablation does not lead to invasion of peripheral macrophages[16], and (3) that glucocorticoids, such as Dex, are established immunosuppressants[53], we interpret the enhanced regenerative effects of dendrimer-Dex conjugates to likely be a result of the enhanced resolution of microglia reactivity. Still, given that MG in species other than fish have been shown to express the glucocorticoid receptor[15] and that glucocorticoids have been shown to have effects on retinal stem cells[15,54], we cannot rule out an additional role for effects of Dex or dendrimer-Dex on MG cells directly. Finally, transcriptomic analysis and functional tests identified a gene, *rnf2*, required for the pro-regenerative effects of post-injury immunosuppression. These findings are in keeping with recent reports showing that microglia/macrophages play context-dependent[18,49,50] and/or species-specific roles during retinal regeneration[10,17], suggesting that immune system modulation will be an important component in promoting neuronal regeneration therapeutically.

Intravital imaging, by accounting for the highly dynamic nature of cellular behaviors and cell-cell interactions in vivo, has revealed key insights into developmental and regenerative

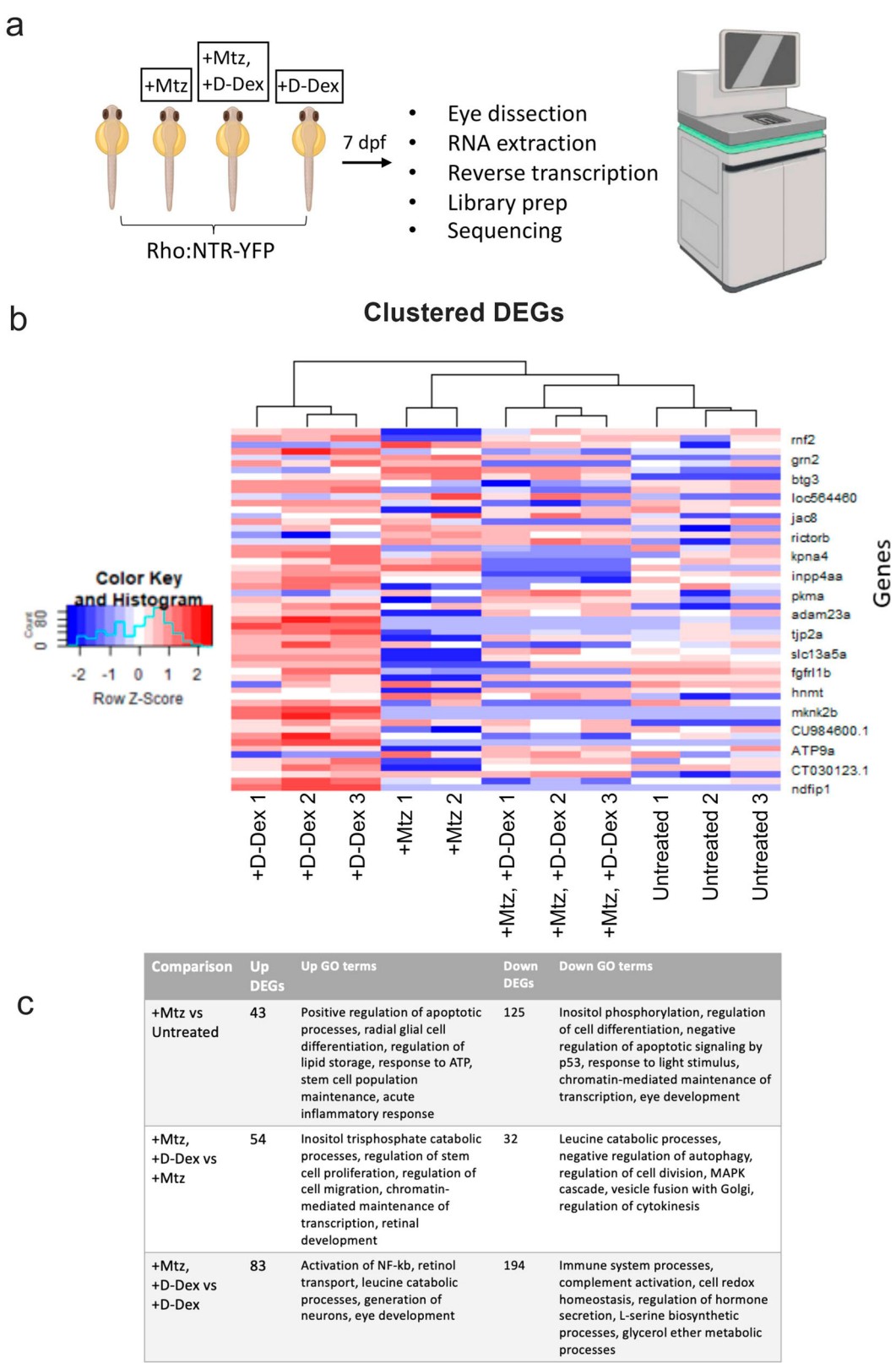

**Fig. 5 RNA-seq identifies differentially expressed genes following D-Dex treatments. a** Assay schematic for RNA-seq assays. Eyes from NTR:rod larvae across four conditions (untreated, Mtz only, +Mtz, +D-Dex, or D-Dex only) were collected at 7 dpf, 24 h following D-Dex injections (where applicable) and processed for RNA sequencing. **b** Subset of most statistically significant DEGs across conditions (red indicates upregulation, blue indicates downregulation). **c** Number of up and downregulated DEGs associated with select GO terms. Fish icons in panel (**a**) were produced with permission from Biorender.

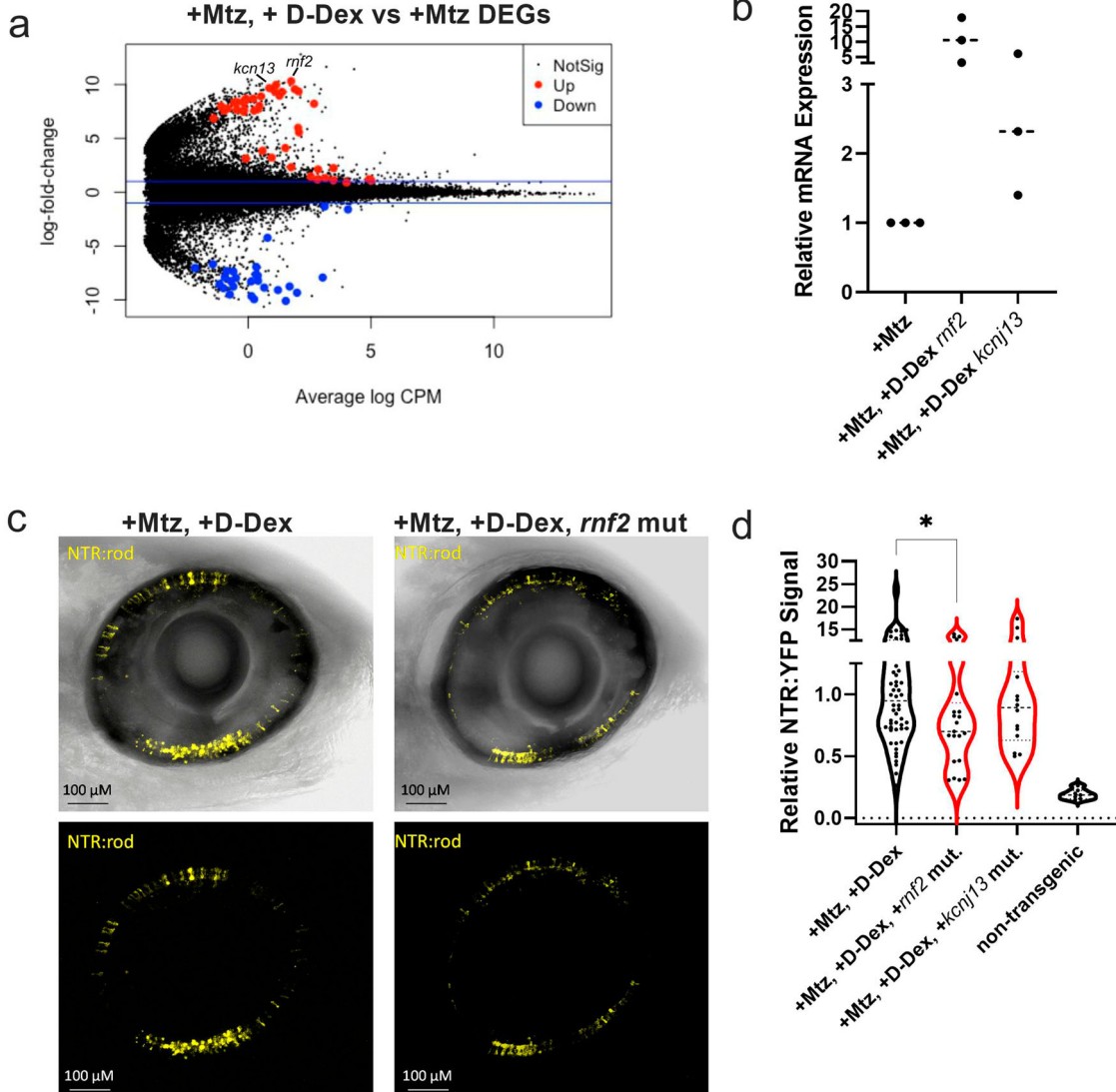

**Fig. 6 The E3 ubiquitin ligase *rnf2* is required for D-Dex enhanced regeneration. a** Volcano plot of DEGs between Mtz only and +Mtz, +D-Dex treated eyes. Red indicates statistically significant upregulation and blue indicates statistically significant downregulation. **b** qRT-PCR assessment of *rnf2* and *kcnj13* upregulation in the +Mtz, +D-Dex condition. **c** Quantification of NTR-YFP-expressing rod cells at 9 dpf to assess changes in regeneration kinetics in +Mtz, +D-Dex larvae following knockdown of *rnf2* and *kcnj13*, n = 54, 22, 24, and 7 from left to right (*$p \leq 0.05$). **d** Representative in vivo confocal images of NTR-rod larval retinas at 9 dpf following +Mtz, +D-Dex treatments in either wildtype control (left) or *rnf2* knockdown "crispant" backgrounds. Sample sizes from left to right: 54, 21, 14, and 7.

processes[55,56]. In our prior study, in vivo time-lapse imaging allowed us to delineate microglia from peripheral macrophage responses to retinal cell loss, surmounting a key limitation of the field[16]. Here, AO-LLSM enabled improved spatiotemporal resolution of microglia dynamics during retinal regeneration. The data showed that post-ablation Dex treatments decreased microglia migration speed, suggesting inhibition of reactivity. In addition, we noted that microglia proximal to ongoing rod cell loss exhibited heterogeneous responses (Supp Movie 3). One subset remained in direct contact with dying rod cells throughout the imaging period, while other cells exhibited highly motile or non-reactive homeostatic behaviors. This data is consistent with evidence of functionally delineable microglia subtypes, suggested by heterogeneous single-cell transcriptomic studies[48,57] and more recently confirmed in mouse models of retinal degeneration[58], in the zebrafish brain[59] and spinal cord[60], and during muscle regeneration in zebrafish[55]. Parallels between our data and the latter study are particularly intriguing, as Ratnayake et al., showed

that "dwelling" macrophages—those remaining associated with the injury site—secreted factors promoting stem cell proliferation. Additional studies will be required to define microglia subtype functions during retinal regeneration, including roles in stimulating MG proliferation.

Clinical trials employing systemic glucocorticoid immunosuppression as a strategy for ameliorating neurodegenerative diseases have failed[20]. Targeted delivery approaches, such as PAMAM dendrimers[22,23,31,61], could be used to reduce undesirable systemic effects and thus harness the benefits of this powerful family of drugs in CNS disorders. We used dendrimers to test the effects of targeted delivery of Dex to reactive microglia on retinal regeneration. The results showed D-Dex formulations reduced Dex-associated toxicity, facilitated the targeting of reactive macrophages, and improved the pro-regenerative effects of post-injury Dex treatments. These findings and related studies showing immunosuppression improves regenerative capacity in the mouse retina[17], suggest that targeted modulation of immune cell

responses to neuronal cell death may represent a generalizable strategy for enhancing neural stem cell activity. Additional studies will be required to determine how accelerated retinal cell regeneration kinetics correlates to the recovery of visual deficits- if the effects of Dex are mediated solely through modification of microglia reactivity—and the impact of targeted immunomodulation in mammalian retinal regeneration models.

To explore molecular mechanisms, we used bulk RNA-seq and qRT-PCR to assess gene expression changes associated with D-Dex-enhanced retinal regeneration and rapid CRISPR/Cas9-enabling knockdown methods to test gene function. D-Dex was associated with gene networks involved in the dynamic regulation of chromatin regulation as well as MAPK and NF-kB signaling, all pathways known to alter MG responses to injury in zebrafish. MAPK and NF-kB signaling pathways are activated upon retinal cell death leading to a pro-inflammatory cytokine cascade that converges on upregulating pro-regenerative genes such as stat3[45,62,63]—our qRT-PCR showed the inflammatory cytokine factor il6st and stat3 were upregulated by D-Dex as well. We identified kcnj13 and rnf2 as being highly upregulated by D-Dex and showed that rnf2, but not kcnj13, was required for the improved pro-regenerative effects of D-Dex. As a key component of the Polycomb repressor complex 1 (PRC1)[64], rnf2 acts as an E3 ubiquitin ligase. As rnf2 has been shown to bind the core activation complex of Notch[65], presumably acting as a repressor of Notch signaling, upregulation of rnf2 by D-Dex may serve to enhance the repression of Notch. As Notch inhibition has been shown to promote retinal regeneration[66,67], knockdown of rnf2 may disrupt D-Dex mediated repression of Notch. As detailed in a recent review, further examination of factors impinging on Notch signaling during retinal regeneration is needed to clarify potential context-specific roles[68].

Collectively, our findings add to a growing appreciation of the importance and context-dependent nature of neuroimmune interactions, underscoring the need for comprehensive interrogations of microglia subtype function during neuronal degeneration and regeneration, as well as the potential value of nanoparticle-based strategies for targeting reactive immune cells. Based on these and related findings, we posit that gaining control of microglia responses to neuronal cell loss may provide a therapeutic strategy for promoting neural repair in humans.

## Methods

**Zebrafish husbandry and transgenic lines**. All studies were carried out in accordance with recommendations by the Office of Laboratory Animal Welfare (OLAW) for zebrafish studies and an approved Johns Hopkins University Animal Care and Use Committee animal protocol. All fish were maintained using established conditions at 28.5 °C with a 14:10 h light:dark cycle. Previously published transgenic lines used here include: Tg(rho:YFP Eco. NfsB)gmc500[35], Tg(mfap4.1:Tomato-CAAX)xt6[69], and Tg(mpeg1.1:LOX2272-LOXP-tdTomato-LOX2272-Cerulean-LOXP-EYFP)w201[70]. Tg(rho:YFP Eco. NfsB)gmc500 expresses a bacterial Nitroreductase (NTR) enzyme selectively in rod cells to enable selective ablation (see below). This line was further propagated in a pigmentation mutant, roy[a9] (roy), to facilitate the detection of the YFP signal in vivo.

**NTR-Mtz mediated rod photoreceptor ablation and free Dex treatment**. NTR:YFP-expressing larvae were separated into equal-sized groups at 5 dpf: (i) nontreated controls and (ii) larvae treated with either 10 mM Metronidazole (Mtz, Acros) for 24 h or 2.5 mM Mtz for 48 h at 5 dpf (specifically to slow down the injury process for in vivo confocal imaging). Mtz is reduced by NTR into a cytotoxic substance inducing DNA damage followed by specific cell death. After Mtz treatment, fish were rinsed and kept in ~0.3 Å Danieau's solution until quantified, imaged, or killed. For experiments with non-conjugated Dex treatment, equal size groups of fish ablated with Mtz were treated with 2.5 μM Free Dex. Fish treated with Dex only as controls were additionally used as well as untreated fish.

**TUNEL staining**. TUNEL staining was performed at 24 h following the onset of Mtz at 5 dpf using the TMR red In Situ Cell Death Detection Kit (Sigma-Aldrich). Following histological preparation and sectioning with a previously published method[16], slides were treated with PBS containing 1% sodium citrate/ 1% TritonX-100 at 4 °C, were rinsed with PBS, and then incubated with TUNEL reaction cocktail for 30 min at 37 °C. Confocal z-stack images were then taken for each slide using a confocal microscope using established protocols[71]. TUNEL+ cells were then counted across 10 control non-ablated and +Mtz retinas.

**Adaptive optics lattice light-sheet microscopy (AO-LLSM)**. About 5–7 dpf zebrafish were anesthetized using tricaine (0.16 mg/ml) and then embedded in a 0.8% low melt agarose droplet on a 25-mm coverslip. A homemade hair loop was used to position and orient fish. The excitation and detection objectives, along with the 25-mm coverslip, were immersed in ~40 ml of E3 media + PTU at room temperature (22 ± 1 °C). Zebrafish expressing Tg(rho:YFP Eco. NfsB)gmc500 and Tg(mfap4.1:Tomato-CAAX)xt6 to label rod photoreceptors and microglia/macrophages, respectively[16]—were excited using 514 nm and 560 nm lasers simultaneously (514 nm operating at ~3 mW and 560 nm operating at 5 mW corresponding to ~15 and ~25 μW at the back aperture of the excitation objective) with an exposure time of 20–50 msec. Dithering lattice light-sheet patterns with an inner/outer numerical aperture of 0.36/0.4 for both colors were used. The optical sections were collected by scanning the sample stage with 400–500 nm step size, equivalent to 200–250 nm axial step size in detection objective coordinate, with a total of 121–201 steps. Emissions from the two fluorophores were separated by a long-pass filter (Di03-R561-t3, Semrock) and captured by two Hamamatsu ORCA-Flash 4.0 sCMOS cameras (Hamamatsu Photonics). Prior to the acquisition of the time series data consisting of 100–200 time points, the imaged volume was corrected for optical aberrations using a two-photon "guide star" based adaptive optics method (manuscript in preparation). Each imaged volume was deskewed and deconvolved with experimentally measured point spread functions obtained from 100 nm tetraspec beads (Thermo Fisher) in C++ using the Richardson–Lucy algorithm on HHMI Janelia Research Campus' computing cluster. The AO-LLSM was operated using custom LabVIEW software (National Instruments).

**IMARIS quantification of microglia dynamics and fluorophore colocalization**. To process images taken with AO-LLSM, deconvolved files were first loaded into Fiji (ImageJ v1.52p; NIH). Each transgene was set to the appropriate color (yellow for 514 nm, cyan for 445 nm, and red for 560 nm) using a Fiji macro (.ijm file) before each channel was merged together in one file and saved as a.tiff file and then a.avi movie file (compression = JPEG, 7 frames per second). Next, each.tiff file encompassing a time series for each image was loaded into IMARIS (v9.5.1, 9.6.1, and 9.7.0; Bitplane) for 4D-rendering. To quantify immune cell dynamics, 15–19 individual microglia (CFP channel) for groups of 5–6 fish were processed for each relevant condition. For these movies, larvae were imaged at 30–180 s intervals and at a volume of 30 microns. In the CFP channel, a surface was created with identical processing parameters, including thresholding to set the fluorescent intensity at a level that properly labeled visible microglia. After processing, automatically quantified data from IMARIS was exported as a.xlsx file, where values for each image were averaged for migration speed, sphericity, and displacement.

To measure the level of colocalization between D-Cy5 and transgenic microglia, the colocalization tool IMARIS was used to identify the number of colocalized pixels between the imaging channel for microglia and D-Cy5. Results were then normalized to the control, non-ablated larvae for each pair of the four pairs of larvae analyzed.

**Synthesis of fluorescently labeled dendrimer (D-Cy5 and D-FITC) and dendrimer-dexamethasone (D-Dex) conjugates**. To enable imaging of dendrimer in this zebrafish model, we fluorescently labeled the hydroxyl-terminated generation 4 PAMAM dendrimers (G4-OH) by covalently conjugating a near IR dye Cyanine 5 (Cy5) to the surface -OH groups of the dendrimers to obtain D-Cy5. These protocols have been previously published[72,73]. The D-Cy5 conjugate was characterized for its loading and purity using proton-nuclear magnetic resonance ($^1$H NMR) and high-performance liquid chromatography (HPLC) respectively. Similarly, Fluorescein isothiocyanate (FITC)-labeled dendrimer (D-FITC) was used for some biodistribution studies. D-FITC synthesis and characterization were performed as described previously. Dendrimer-dexamethasone conjugate (D-Dex) was synthesized by following a previously reported synthesis procedure[42]. Briefly, the D-Dex conjugates were synthesized using a two-step procedure. In the first step, dexamethasone-21- glutarate (Dex-linker) was synthesized by reacting the -OH group at the 21$^{st}$ position of Dex to the -COOH of the glutaric acid in the presence of triethylamine as a base. The Dex-Linker was purified using flash column chromatography. In the second step, the Dex-Linker was conjugated to the -OH groups on the dendrimer surface in the presence of coupling agent benzotriazol-1-yloxy tripyrrolidinophosphonium hexafluorophosphate diisopropylethylamine (PyBOP) as a base and anhydrous dimethylformamide (DMF) as solvent. The final product was purified using dialysis against DMF to remove reacted Dex-linker and PyBOP side products for 24 h, followed by dialysis against water to remove DMF. D-Dex conjugate was characterized using $^1$H NMR to estimate the drug loading and HLPC for its purity.

**Intravital confocal microscopy**. All intravital imaging applied previously detailed protocols[71]. Fish expressing YFP and NTR in rod photoreceptors, Tg(rho:YFP Eco. NfsB)gmc500, and tdTomato in microglia/macrophages, Tg(mpeg1.1:LOX2272-LOXP-tdTomato-LOX2272-Cerulean-LOXP-EYFP)w201, were used. Confocal z-stacks encompassing the entire orbit of the eye or entire fish (step size, 5 microns)

were collected at 10- or 20-min intervals over a total of between 4.5 and 20 h. Image analysis was performed using Fiji (i.e., ImageJ v1.49b; NIH) or Imaris (v7.6.5; Bitplane) to quantify by morphometric analysis as sphericity, a correlation between area and volume or by volume in the kidney using minimum bounding spheres. All in vivo confocal images were imaged using a 20x objective, with all histological images imaged using a 40x objective.

**IMARIS volumetric rendering and quantification.** Larval whole-retina images were collected as described above using identical acquisition and IMARIS processing parameters across all conditions. Photoreceptor volume was calculated using IMARIS local background-based volumetric rendering of YFP signals.

**ARQiv scans to measure rod photoreceptor regeneration kinetics.** *Tg(rho:YFP Eco. NfsB)gmc500* larvae were treated with Mtz ± Dex, and regeneration kinetics were analyzed using the ARQiv system, as previously described in ref. [16]. About 5 dpf larvae were exposed to 10 mM Mtz for 24 h and then were treated with a single dose of 2.5 μM Free Dex or treated with D-Dex AT 6 dpf following visual confirmation of loss of YFP-expressing rod cells using stereo-fluorescence microscopy. At 7 dpf all fish were placed in fresh media until the ARQiv scan was performed on day 9.

**Histology labeling of dividing cells with proliferative cell nuclear antigen (PCNA) Bromodeoxyuridine (BrdU).** To label dividing cells with antibodies for PCNA at 8 dpf or incorporated BrdU at 10 dpf, YFP-NTR larvae were ablated with Mtz 5 dpf for 24 h, and half then received a pericardial injection of D-Dex at 6 dpf. Following sacrifice and histological preparations of sections as previously published in ref. [16], the primary antibody anti-pcna (mouse, clone PC10, Sigma-Aldrich) or anti-BrdU (mouse, clone BU-33, Sigma-Aldrich) was used in conjunction with Alexa Fluor 635 secondary antibody (Molecular Probes). Slides were then imaged as z-stacks with confocal microscopy and pcna/BrdU+ cells inside the neural retina (excluding the CMZ and lens/endothelial cells) were counted and compared between +Mtz only and +Mtz and D-Dex retinas.

**Pericardial injection of Dex and dendrimer-conjugated Dex and D-Cy5.** *Tg(rho:YFP Eco. NfsB)gmc500* larvae were treated with Mtz at 5 dpf, and then at 6 dpf larvae were anesthetized with Tricaine, placed under a PLI-100 picospritzer (Harvard Apparatus) and injected with 10 nL amount of either 5 μM Dex, D-Dex or D-Cy5. The concentration was doubled compared to Free Dex to account for differential uptake under injection compared to soaking. Subsequently, at 9 dpf, the ARQiv scan was performed to measure regeneration kinetics as above. In later experiments, the injection was performed with D-Cy5 in order to observe Dendrimer activity in vivo. The toxicity of this injection method was determined following two trials of injection of equal groups of 12 fish with vehicle, and various concentrations of Free Dex, or D-Dex (containing equivalent Dex in conjugated form) at 5 dpf. A total of ~500 fish were used across the two timepoints. At 7 dpf, % of surviving fish was counted (Fig. 2) and LD$_{50}$ was calculated using previously established methods[35].

**Statistics and reproducibility.** Data were processed using GraphPad Prism 9 to generate all plots as well as carry out statistical analyses recommended by the software based on the type of data generated. For tests involving only two experimental groups (example: Supplementary Fig. 1c comparing TUNEL+ cell staining), unpaired two-tailed *t*-tests were conducted to identify *p* values and effect sizes between groups. The majority of assays involved comparisons across multiple groups (example: Fig. 1e), and statistical tests done in these cases were unpaired two-tailed Welch's ANOVA *t*-tests and included a multiple comparisons correction of *p* values. Prism was also used to calculate 95% confidence intervals were calculated for the toxicity curve (Fig. 2b) using the Kaplan–Meier method. All in vivo experiments were conducted using a minimum of three biological replicates (sibling transgenic fish maintained in separated media along with drug, where applicable), including each control and treatment condition. At all larval stages where experiments were conducted, sex determination has not yet occurred and thus was not considered in the experimental setup as it would be with adult fish.

**Bulk RNA sequencing—experimental design, RNA extraction, and sequencing.** For RNA sequencing, sibling 5 dpf gmc500 fish were screened for YFP expression and split into 12 groups (15 larvae in each sample with three biological replicates each for four conditions: Mtz only, Mtz + D-Dex injection, No Mtz, or No Mtz + D-Dex injection). At 7 dpf, 24 h into recovery from 24 h soaking with 10 mM Mtz (for applicable groups) and 24 h after pericardial injection of 10 nL of 2.5 μM D-Dex (for applicable groups), whole eyes were dissected from each larvae. Dissected eyes were placed into Trizol (ThermoFisher Scientific, 15596018) and RNA was extracted using the RNeasy Micro Kit (Qiagen, 74004). Samples were then sent to the Deep Sequencing and Microarray Core (Johns Hopkins University) for library preparations and sequencing. Samples were processed using the ClonTech SMARTer Ultra Low Input RNA-v4 (Takara 634440) kit, following the manufacturer's instructions. Briefly, cDNA is generated from 10 ng total RNA using oligo dT primer. cDNA was amplified and purified. About 150 pg was taken into the Illumina Nextera XT DNA (Illumina FC-131-1096) library prep kit. Tagmentation was performed and libraries were amplified using Illumina dual indexed

adapter mix. Samples were pooled and sequenced on the NovaSeq 6000 SP 100 flow cell, single end, 100 bp reads, yielding ~50 M reads/sample.

**Bulk RNA-sequencing analysis.** Raw sequencing data (in the form of fasta.gz files) was downloaded and unzipped, and then terminal function cutadapt was used to remove Nextera sequencing adapters (specifically, the sequence CTGTCTCTTATA was trimmed). Samples were then read into fastqc for quality control to ensure proper and similarly sized libraries, complete removal of adapters, and high sequence quality scores throughout. Sequencing for each sample occurred over two lanes and thus read files were combined using the "cat" terminal function. Read files were then mapped to the most recent Ensembl reference genome for Danio rerio GRCz11/danRer11 using kallisto. The percentage of uniquely mapped reads was identified by fastqc and ranged from 69–75% for the 15 samples.

To identify differentially expressed genes (DEGs) between conditions, a matrix file containing all Ensembl transcript ID's and 12 columns, one for each sample, were read into the edgeR Bioconductor package (version 3.34.1) in R/R Studio (versions 4.0.3 and 1.4.1103, respectively)[74]. Briefly, a model matrix was designed to aggregate data between three biological replicates for each condition. Further analysis of the variance between samples revealed low-quality data for one replicate of Mtz-only RNA compared to the other groups, and thus this group was removed. Next, all possible pairwise comparisons between the 4 samples were completed using the glmLRT method. Data were then exported as.csv files. Prior to identifying hit DEGs, control sample comparisons were used to filter out transcripts for comparisons of interest. Finally, the remaining transcripts were sorted to identify hit DEGs that demonstrated at least a log fold change in expression of 1 (an expression fold change of 2) in either direction and with a false discover rate (FDR) <0.10.

**Functional annotation of DEGs and pathways.** To identify broader functional changes among each condition, Gene Ontology (GO, http://geneontology.org/docs/go-enrichment-analysis/) was used. For relevant pairs of conditions, transcript IDs for hit genes were entered into BioMart to yield a list of gene names that were then put through GO.

**Quantitative real-time polymerase chain reaction (qRT-PCR).** To validate gene expression changes identified in RNA-sequencing, a select few genes of interest were identified for qRT-PCR analysis. YFP-NTR expressing larvae at 5 dpf were screened and ablated with Mtz (10 mM, 24 h) as normal and 24 h after removal at 5 dpf whole eyes were dissected for ~16 larvae from each condition and pooled for mRNA extraction. A previously published protocol was used to measure gene expression with qRT-PCR[36]. Briefly, RNA was purified using NEB Monarch RNA Cleanup kit and reverse transcribed to cDNA using qScript cDNA synthesis kit (QuantaBio). Quantitative PCR was conducted using designed primers with the primerdb database and PowerUp™ SYBR™ Green Master Mix (ABI) in QuantaStudio (ABI). Delta delta CT analysis was performed to calculate the relative fold change in gene expression levels between knockdown larvae and controls. Each experiment was performed in triplicate.

**CRISPR/Cas9 redundant gene targeting.** Based on bulk RNA-sequencing data (specifically, upregulation in Mtz+D-Dex compared to Mtz alone), the genes *kcnj13,rnf, col11a2,* and *nosip* were selected for follow-up functional testing. For *kcnj13* and *rnf2,* we used ZFIN (zfin.org) to confirm expression in the eye as well as known roles in immune system function. CRISPR/Cas9 mediated redundant targeting injections were performed at the one-cell stage of gmc500 embryos utilizing the published strategy by ref. [47]. The four published sgRNAs for each gene was ordered as DNA oligos (Supplementary Data 1), assembled with the general CRISPR tracr oligo, and then transcribed using pooled in vitro transcription (HiScribe T7 High Yield RNA Synthesis kit, New England BioLabs) and cleaned up with the NEB Monarch RNA Cleanup kit. A mixture of four sgRNAs (1 ng in total) and Cas9 protein (2.5 μM, IDT) was injected into *rho:YFP-NTR* embryos at the one-cell stage for targeting each gene. Mutant larvae were then taken through the same regeneration assay ending at 9 dpf as above.

**Reporting summary.** Further information on research design is available in the Nature Portfolio Reporting Summary linked to this article.

# Data availability

RNA sequencing data that support the findings of this study have been deposited in GEO with the accession code GSE216060. Data can be downloaded by going to ncbi.nlm.nih.gov/geo and searching for the accession code. The source data for the main figures are given in Supplementary Data 1 and 2, and any remaining information can be obtained from the corresponding author upon reasonable request.

# Code availability

The specific code used in the analysis of bulk RNA-sequencing data is available upon request to authors.

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

## Acknowledgements

This work was supported by the following grants from the National Institutes of Health, R01OD020376 (J.S.M.), R01EY025304 (R.M.K.), F31EY032790 (K.E.), and P30EY001765-45 (core grant to Wilmer Eye Institute). We thank the Ramakrishnan and Tobin laboratories for sharing transgenic fish and members of the Mumm lab for providing helpful discussions. We also thank BioRender.com for the permission to use scientific icons in making figures.

## Author contributions

J.S.M., K.E., and D.T.W. designed the project. K.E. and D.T.W. carried out all experiments in zebrafish and, along with J.S.M., wrote the manuscript with input from all authors. S.P.K. and R.M.K. oversaw the development of nanoparticle-conjugated drugs and contributed to the design of related experiments. T.-M.F. and E.B. oversaw and carried out AO-LLSM imaging experiments, along with J.S.M. and K.E. K.E. carried out RNA sequencing and analysis, with assistance in analysis from Z.C., S.L.W., and S.N. N.K., G.L.C., and A.S. assisted with confocal microscopy imaging/analysis along with K.E. and D.T.W. M.T.S. contributed to manuscript writing and editing.

## Competing interests

J.S.M. holds patents for the NTR inducible cell ablation system (US #7,514,595) and uses thereof (US #8,071,838 and US#8431768). R.M.K. and S.P.K. have been awarded pending patents relating to the hydroxyl dendrimer platform for ocular therapies. RMK and his wife (Sujatha Kannan) are co-founders/board members and have financial interests in Ashvattha Therapeutics Inc., a start-up focusing on clinical translation of the dendrimer platform. The remaining authors declare no competing interests.
