## [Peer Review File · Communications Biology]

Nanoparticle-based targeting of microglia improves the neural regeneration enhancing effects of immunosuppression in the zebrafish retinaReviewers' comments:

Reviewer #1 (Remarks to the Author):

Review

Emmerich et al analyse dendrimer-targeted immunosuppression of microglia in a zebrafish model of photoreceptor regeneration. They apply intravital time lapse imaging to assess the impact of Dexamethasone on microglia post ablation and test the impact of dendrimer-conjugated dexamethasone on regeneration. While this is clearly an interesting approach, their data is unfortunately not presented in a way that is sufficient to draw conclusions.

The authors state the time scale for scanning intervals (5 to 1880 sec/interval) – not clear how this is relevant here – morphology and speed read outs only. These time intervals would allow to assess microglia process speed for example – however no data in this direction are provided.

When mentioning different tg lines for the first time in the results it's important to give full names, so the reader knows exactly which lines have been used.

I appreciate that authors have published the ablation assay before – nevertheless data should be shown to proof ablation of rod cells in this study.

The pure impact on MTZ on microglial morphology would be interesting – such a control should be included.

Measurements of Sphericity and Migration Speed: the authors state that 6-8 larvae have been used – no statement on how many cells have been analysed. Furthermore, quantifying microglial morphology is not trivial – more details on the method used would be needed here.

Conclusions on the peripheral macrophage migrating inside the retina should be backed up by using additional markers.

Toxicity studies: how many fish used here? Why are no statistics provided?

D-Cy5 localisation studies: images are not convincing. Single channels should be provided in addition. Co-localisation should be quantified. Is the localisation of D-Cy5 specific? Would other particles show similar localisations?

Reviewer #2 (Remarks to the Author):

The paper by Emmerich and colleagues investigates how dexamethasone influences the reactivity of retinal microglia and how conjugation of Dex to novel nanoparticles influence the regeneration of photoreceptors in larval zebrafish. The authors apply cutting-edge imaging techniques (AO-LLSM) to study retinal regeneration in larval zebrafish. Further, the authors apply PAMA dendrimer nanoparticles to target immune cells in the retina. Although the technology applied and some of the

imaging data are pretty cool, the main claims of the paper, namely selective targeting of microglia and super-accelerated retinal regeneration, are not robustly supported by data.

The author should consider the following points:

- The appropriate control for the D-Dex would be the dendrimer alone. It is possible that dendrimer alone influences the inflammatory state of immune cells in the eye and perhaps even the reprogramming of Muller glia into damaged induced retinal stem cells,
- The paper would benefit from additional mechanistic investigations. For example, does the D-Dex directly impact cell-signaling in the microglia? Do the microglia express receptors for Dex? The localization of the Cy5-dendrimer in Figure 4 does not convincingly argue that the site of action is microglia in areas of regenerating retina. To the contrary, these data suggest that the dendrimer predominantly accumulates at extra-retinal sites, which could secondarily influence the regeneration of photoreceptors and also influence retinal glia.
- The authors frame the study with relevance to treating human disease by selectively targeting anti-inflammatory drugs to retinal microglia to avoid side-effects. This is, indeed, a noble cause and worthy of investigation. However, the model and mode of application are very far removed from a pre-clinical model. The ideal model would involve applying D-Dex, free Dex vs unlabeled dendrimer via eye drops (or intravitreal injection) to a mammalian disease model involving ocular inflammation. I think the paper should be re-framed.
- The authors claim immunosuppression of microglia. However, the only significant difference in phenotype resulting from treatment with Dex is the rate of migration. This is a bit thin. Ideally data would be provided for levels of pro/anti-inflammatory cytokines or changes in expression of genes associated with reactive microglia.

Minor points:

Figure 3 – it would be nice to have detailed histological data to go with the gross measures of fluorescence from NTR-YFP.

“super-accelerated regeneration” is over-stating the outcome.

Reviewer #3 (Remarks to the Author):

This manuscript describes a study investigating the role that injury-induced inflammation plays in governing the regeneration of photoreceptors in zebrafish. Significantly, this manuscript also describes and validates the delivery of dendrimer-conjugated dexamethasone for microglia-specific, targeted immune suppression and a novel microscopy technique for imaging live zebrafish larvae. The technical innovations described in this manuscript are solid and will make an important contribution to the field. The underlying cause of the accelerated rod regeneration following DEX treatment is subject to multiple interpretations, which the authors should strive to include in revisions made to this manuscript.

Below are (mostly minor) points the authors should consider incorporating into any revisions of this manuscript.

Abstract

Line 48: As written, this sentence states that ‘insights’ ‘stimulate limited regenerative responses.’ This sentence should be rewritten for clarity.

Introduction

Line 88: References should be added to support this sentence. This will aid readers, who may be grappling with this issue.

Results

Line 115: ‘Post-ablation treatment suppresses microglia reactivity....’ Actually, post-ablation Dex treatment alters the motility, and presumably also phagocytosis, of microglia. By virtue of the marked

change from ramified to amoeboid morphology, it is clear that microglia in the Dex-treated retinas sense the death of the rod cells and are reactive. The authors may wish to be more circumspect regarding the conclusions (and interpretations) made from the cellular data.

Line 153: 'Notably, ...' I do not believe the authors provide sufficient evidence to state unequivocally that peripheral macrophages enter the eye when rod cells die. In the adult zebrafish retina, in contrast to mice, microglia reside among cells of the RPE. It is possible that some microglia migrating into the outer nuclear layer originate from above this layer, not outside the retina (is it even possible for macrophages to traverse Bruch's membrane?). The authors state in the legend to Supplemental Figure 1 that these microglia may migrate from the RPE. Microglia originating from the RPE layer is the most parsimonious interpretation of the images and videos and should be stated as such in the Results.

Line 156: Data is the plural of datum. Therefore, 'These data clarify....'

Discussion

Following photoreceptor death in zebrafish, pro-inflammatory molecules originate from dying photoreceptors, Müller glia and microglia. Dendrimer-based targeting of dexamethasone to microglia may not impact the synthesis and release of pro-inflammatory molecules originating from cellular sources other than microglia, or even from microglia, themselves. The presence (and perhaps persistence) of pro-inflammatory molecules, even when Dex is present, may alter the kinetics of regeneration.

Dex and D-Dex treatments after rod cell death suppress microglial migration and phagocytosis. Phagocytosis, both by altering gene expression in macrophages and removing cellular debris, is a known mechanism for resolving inflammation. The absence of phagocytosis and/or persistence of photoreceptor corpses may alter the inflammatory environment within the outer retina in a manner that may accelerate the kinetics of rod regeneration.

Do the Dex treatments after rod cell death suppress inflammation in a manner similar to systemic pre-treatment with Dex, or do these treatments merely disable microglial migration? The authors conclude (line 248) that decreased migration speed is evidence of 'an immunosuppressive effect.' Are there data that support this statement? Are alternative interpretations possible?

To be complete, the authors may wish to discuss the various sources of pro-inflammatory molecules in the zebrafish retina, the complexities underlying the inflammatory response during photoreceptor death, microglial phagocytosis and neuronal regeneration and the possibility that disabling migration of microglia may result in shifting the inflammatory milieu within the outer retina in a manner that is sufficient to accelerate rod regeneration.

Methods

Line 343: PTU is not without side effects. The authors may wish to determine whether or not PTU has any impact on the innate immune system.

Figures

In the photomicrographs of the outer nuclear layer, it appears that outer retina is on the left and inner retina is to the right, though this is only a guess. The orientation of the retina within the photomicrographs (and videos, if possible) should be made explicit.

Reviewers' comments:

Reviewer #1 (Remarks to the Author):

Review

Emmerich et al analyse dendrimer-targeted immunosuppression of microglia in a zebrafish model of photoreceptor regeneration. They apply intravital time lapse imaging to assess the impact of Dexamethasone on microglia post ablation and test the impact of dendrimer-conjugated dexamethasone on regeneration. While this is clearly an interesting approach, their data is unfortunately not presented in a way that is sufficient to draw conclusions.

Comment 1: The authors state the time scale for scanning intervals (5 to 1880 sec/interval) – not clear how this is relevant here – morphology and speed read outs only. These time intervals would allow to assess microglia process speed for example – however no data in this direction are provided. **The Methods have been updated with only the scan intervals used for each experimental section indicated (line 170). We have eliminated all references to scan times that were not included in the analysis here, limiting the range to 30-180 second intervals. The full range of intervals collected were provided to simply showcase advantages of the imaging technology from Dr. Betzig's lab, i.e., improved temporal resolution of microglia dynamics. We focused our analysis on metrics previously linked to microglia reactivity, e.g., migration speed and sphericity. To our knowledge, although certainly interesting, process speed has not been used as a measure of macrophage reactivity. Moreover, given the dynamic nature of the processes we observed, this would have required a series of studies at the 5-15 second range. Unfortunately, COVID restrictions disrupted our ability to perform additional experiments and we had to focus on assays facilitating known measures of macrophage reactivity in the interest of time.**

Comment 2: When mentioning different tg lines for the first time in the results it's important to give full names, so the reader knows exactly which lines have been used. **We thank the reviewer for this reminder, all transgene and allele numbers are now provided at first mention within the Results (lines 136, 176, and 290).**

Comment 3: I appreciate that authors have published the ablation assay before – nevertheless data should be shown to proof ablation of rod cells in this study.

We appreciate the need to explore the robustness of all methods used, including the nitroreductase/metronidazole selective cell ablation technology that Dr. Mumm developed. Thus, we have included a new supplemental figure (Supp. Figure 1) to demonstrate the loss and regeneration of NTR:YFP using *in vivo* imaging and included TUNEL staining as a marker of DNA damage (see new figure below). In a recent publication using the same transgenic line, we have characterized the specific type of cell death elicited by Mtz exposure in NTR-YFP expressing rod cells (Zhang et al.; eLife, 2021). We refer readers to the eLife paper for further details of this important question.

Comment 4: The pure impact on MTZ on microglial morphology would be interesting – such a control should be included.

We thank the reviewer for the suggestion of adding this control to our study. Metronidazole is an antibiotic thus assessing any effects on immune system reactivity is important. While numerous publications come out every year utilizing this antibiotic to study a variety of biological phenomena, investigations into the effect of the drug alone are lacking. Notably, a recently published thesis from Uppsala University, entitled “Assessment of zebrafish embryo toxicity of environmentally relevant antibiotics” by Ophelia Mastrangeli, investigated the effects of Mtz and a host of other similar antibiotics on the development and behavior of zebrafish. Broadly, they found a lack of significant differences among these metrics when Mtz was administered. To test for effects of Mtz on microglia specifically, we evaluated microglia morphology and behavior in the absence of cell ablation (i.e., in fish not expressing NTR) using *in vivo* time-lapse imaging of reporter-labeled microglia (Supp. Fig. 3). Qualitatively, we saw no changes in microglia or macrophage morphology in the presence of Mtz across multiple regions of the fish (the eye and brain primarily). Comparisons between Mtz-treated and non-treated controls showed no significant differences in the quantifiable metrics we use to assess microglia reactivity (displacement, speed and sphericity, Supp. Fig. 3). These data suggest Mtz, in the absence of cell ablation, has no effect on microglia morphology or behavior.

Comment 5: Measurements of Sphericity and Migration Speed: the authors state that 6-8 larvae have been used – no statement on how many cells have been analysed. Furthermore, quantifying microglial morphology is not trivial – more details on the method used would be needed here.

We now include information regarding the total number of cells analyzed for these measurements (66 microglia, line 183) and included more details on the methodology used (lines 589-607). This information enables those unfamiliar with IMARIS and similar software packages to perform an equivalent level of analysis. Interestingly, we determined that our previous method of averaging sphericity measures across time per cell and per each fish likely limited our ability to detect Dex-associated changes due to highly dynamic nature of morphology changes we observed. Using an alternative method that treated each time point as a separate observation, thus avoiding loss of data due to averaging, Dex-associated reductions in sphericity could be detected. Please see our response to Comment 12 for further details on the application for these measurements.

Comment 6: Conclusions on the peripheral macrophage migrating inside the retina should be backed up by using additional markers.

We thank the reviewer for this comment. After further review, we have concluded that our data is insufficient to come to this conclusion. Accordingly, we have removed discussion of this topic from the manuscript. We nevertheless find this issue to be very important and intend to follow up on this question using additional markers and resources in the future.

Comment 7: Toxicity studies: how many fish used here? Why are no statistics provided?

We have now included additional details regarding the number of fish used in this assay (504 total fish- 24 fish per condition/concentration over 2 trials, line 247). We have now quantified the LD50 (lethal dose causing 50% of the death) for the Dexamethasone soaking group, the only group where any concentration led to death in 50% of larvae as well as performed Chi-square analyses to compare toxicity between conditions (lines 253-258).

Comment 8: D-Cy5 localisation studies: images are not convincing. Single channels should be provided in addition. Co-localisation should be quantified. Is the localisation of D-Cy5 specific? Would other particles show similar localisations?

To address this concern, we have conducted additional imaging experiments to investigate the co-localization of D-Cy5 with microglia and macrophages. Specifically, we expanded our *in vivo* time-lapse imaging to include 4 pairs of larvae that were either treated with 2.5 mM Mtz for 24h prior to imaging (same scheme as the fish from the initial submission) or without Mtz. Importantly, this enabled us to quantify co-localization between D-Cy5 and microglia imaging channels in rod cell ablated and non-ablated control larvae. We observed a significant increase in the number of colocalized pixels when rod cells were ablated. In an updated figure (Fig. 3) we have included additional images to support these data including insets of zoomed in portions of the retina in +Mtz conditions to show microglia/dendrimer interactions as well as highlighting various points in the retinas of fish in both conditions where dendrimer and microglia interactions were observed to occur. We evaluated single channel images as well but do not feel these will help readers to see evidence of co-

localization and so have not included them in the final figure. As we were able to statistically resolve a significant increase in co-localization in ablated fish, we believe we have supported the claim that reactive microglia are targeted by, or increasingly interact with, dendrimers (lines 298-302).

Regarding the question of whether other particles would show similar co-localization, we feel an analysis of additional nanoparticles is beyond the scope of our study. To partially address this issue, we now include a reference showing that SiO₂ nanoparticles show evidence of the opposite trend, i.e., lower interactions with reactive macrophages (lines 302-304). We feel it is also important to note that the Kannan lab has thoroughly characterized PAMAM dendrimers biochemically (Soiberman et al., Biomaterials, 2017) and showed ample evidence of dendrimers targeting activated microglia and macrophages in multiple models of neuroinflammation in a diverse array of species (e.g., Dai et. al., Nanomedicine, 2010; Sharma et. al., Journal of Controlled Release, 2020; Nino et. al., 2018, Science Translational Medicine). Our intent here was merely to confirm dendrimer targeting of microglia in fish.

Reviewer #2 (Remarks to the Author):

The paper by Emmerich and colleagues investigates how dexamethasone influences the reactivity of retinal microglia and how conjugation of Dex to novel nanoparticles influence the regeneration of photoreceptors in larval zebrafish. The authors apply cutting-edge imaging techniques (AO-LLSM) to study retinal regeneration in larval zebrafish. Further, the authors apply PAMA dendrimer nanoparticles to target immune cells in the retina. Although the technology applied and some of the imaging data are pretty cool, the main claims of the paper, namely selective targeting of microglia and super-accelerated retinal regeneration, are not robustly support by data.

The author should consider the following points:

Comment 9: The appropriate control for the D-Dex would be the dendrimer alone. It is possible that dendrimer alone influences the inflammatory state of immune cells in the eye and perhaps even the reprogramming of Muller glia into damaged induced retinal stem cells,

We agree that this control is necessary to demonstrate that the effect of D-Dex enhanced retinal regeneration is specific to the Dexamethasone rather than the dendrimer. We have performed the same regeneration assay using our fluorescence plate readers with injections of dendrimers alone. We did not observe any evidence that dendrimers alone improved the rate of rod photoreceptor regeneration whatsoever. These new results can be found in figure 4b and lines 333-335.

a

b

c

d

e

f

g

Comment 10: The paper would benefit from additional mechanistic investigations. For example, does the D-Dex directly impact cell-signaling in the microglia? Do the microglia express receptors for Dex? The localization of the Cy5-dendrimer in Figure 4 does not convincingly argue that the site of action is microglia in areas of regenerating retina. To the contrary, these data suggest that the dendrimer predominantly accumulates at extra-retinal sites, which could secondarily influence the regeneration of photoreceptors and also influence retinal glia.

We agree that additional mechanistic investigations are needed to demonstrate effects of D-Dex on microglia signaling. However, as direct correlations are difficult *in vivo*, requiring tools for selectively manipulating microglia in zebrafish which are not readily available, we feel such studies are beyond the scope of the current study. We have previously shown that co-ablation of microglia and rod cells disrupts the regenerative process (White et al., PNAS, 2017). This fact precludes evaluations of Dex effects in the absence of microglia. Studies from the Fischer lab have investigated glucocorticoid receptor (GCR) expression, encoded for by the gene *nr3c1*, in the retina of multiple species (mouse, guinea pig, dog and human; Gallina et. al., Development, 2014). These data show that the GCR was primarily expressed in retinal Müller glia rather than microglia. As Dex treatments reduced microglia reactivity, they hypothesized that the suppression of microglia reactivity is secondary to direct effects on MG signaling. Profiling the cross-talk between microglia and MG has become a priority for the Mumm lab and we are planning to use single-cell transcriptomics to address these topics in the future. Our data on Dex effects on retinal regeneration, and confirmation of these effects in several labs subsequently, suggest that Dex will be useful in examining signaling networks mediating enhanced retinal regeneration.

With respect to the comment about D-Cy5 data - we addressed these concerns above in response to reviewer 1 (see comment 8). The presence of the D-Cy5 signal outside of the retina is a result of the pericardial injection method used to introduce D-Dex into fish and subsequently the retina. By performing new imaging experiments in ablated and non-ablated controls, we now show quantified evidence that D-Cy5 co-localization is increased in reactive microglia. Additionally, we added a control to our regeneration assay to show that dendrimer injections alone have no effect on rod cell regeneration kinetics, suggesting that they do not influence the regenerative response through an unknown mechanism (see response to comment 9).

To further investigate the mechanism by which D-Dex acts to enhance retinal regeneration, we performed multiple new experiments using established methods for characterizing regenerative responses including: (1) labeling proliferative Müller glia with anti-PCNA (proliferative cell nuclear antigen; Fig. 4c-d), (2) administering thymidine analog BrdU (Bromodeoxyuridine) using a pulse-chase method (Fig. 4e-g), (3) bulk-RNA sequencing (Fig. 5), and (4) functional tests of DEGs identified in bulk-RNA seq data (Fig. 6). The results section pertaining to figures 4-6 is found in lines 321-459).

a

b

c

Comparison	Up DEGs	Up GO terms	Down DEGs	Down GO terms
+Mtz vs Untreated	43	Positive regulation of apoptotic processes, radial glial cell differentiation, regulation of lipid storage, response to ATP, stem cell population maintenance, acute inflammatory response	125	Inositol phosphorylation, regulation of cell differentiation, negative regulation of apoptotic signaling by p53, response to light stimulus, chromatin-mediated maintenance of transcription, eye development
+Mtz, +D-Dex vs +Mtz	54	Inositol trisphosphate catabolic processes, regulation of stem cell proliferation, regulation of cell migration, chromatin-mediated maintenance of transcription, retinal development	32	Leucine catabolic processes, negative regulation of autophagy, regulation of cell division, MAPK cascade, vesicle fusion with Golgi, regulation of cytokinesis
+Mtz, +D-Dex vs +D-Dex	83	Activation of NF-kb, retinol transport, leucine catabolic processes, generation of neurons, eye development	194	Immune system processes, complement activation, cell redox homeostasis, regulation of hormone secretion, L-serine biosynthetic processes, glycerol ether metabolic processes

Comment 11: The authors frame the study with relevance to treating human disease by selectively targeting anti-inflammatory drugs to retinal microglia to avoid side-effects. This is, indeed, a noble cause and worthy of investigation. However, the model and mode of application are very far removed from a pre-clinical model. The ideal model would involve applying D-Dex, free Dex vs unlabeled dendrimer via eye drops (or intravitreal injection) to a mammalian disease model involving ocular inflammation. I think the paper should be re-framed.

The Kannan lab has previously published experiments similarly to what the reviewer has suggested here (Soiberman et al. 2017; Biomaterials). In our introduction, we reference additional papers to provide background on dendrimer research performed in other mammalian models, and that have advanced to clinical trials. In that context, we view the larval zebrafish model to be novel and complementary, providing unique insights not possible with mammalian systems, for instance: (1) testing the toxicity of free Dex vs. D-Dex in a large sample size (~500 fish, see Fig. 2), (2) assessing effects on retinal regeneration (also at large sample sizes, see Figs. 4 and 6), and (3) enabling *in vivo* imaging detailing immunosuppressive effects of Dex on microglia reactivity and demonstrating the dynamics of dendrimer targeting of microglia.

Additionally, using the same inducible rod cell death model, we recently identified neuroprotective compounds using high-throughput *in vivo* drug discovery; screening over 3,000 drugs in ~350,000 fish larvae. Collaboratively, we went on to show that a subset of these compounds showed neuroprotective effects in mouse photoreceptor cultures and in the *rd1* mouse model of retinitis pigmentosa (Zhang et. al., eLife, 2021). Finally, zebrafish are currently the premier pre-clinical model for the study of retinal regeneration. Importantly, insights derived from fish studies have been used to stimulate regenerative potential in the mouse retina – as we discuss in the manuscript. Accordingly, we feel that zebrafish are a wholly appropriate pre-clinical model for these studies. Moreover, we feel that speculation about therapeutic applications is always just that, speculation, regardless of which animal species is being used as a proxy for human studies. Therefore, we also feel that we are fully warranted in hypothesizing how our findings could inform the development of therapeutic applications. In light of this, we would ask the reviewer to reconsider their position.

Comment 12: The authors claim immunosuppression of microglia. However, the only significant difference in phenotype resulting from treatment with Dex is the rate of migration. This is a bit thin. Ideally data would be provided for levels of pro/anti-inflammatory cytokines or changes in expression of genes associated with reactive microglia.

We agree with the reviewer. To expand our analysis of microglia reactivity, we now include quantifications of 3 parameters used to assess microglia reactivity: average migration speed, average displacement and sphericity (Fig. 1e-g). All three parameters show evidence of Dex-induced suppression of microglia reactivity.

Next, we attempted to address the levels of pro/anti-inflammatory cytokines using qRT-PCR upon treatment with Dex. Unfortunately, we found across multiple trials that treatment with Mtz led to issues with the levels of housekeeping genes used to normalize expression (e.g., *actb1*, *rplp0*). This led us to use bulk RNA-seq to investigate the effects of D-Dex. We elected to test the D-Dex condition for this analysis rather than Dex alone due to increased effect size on regeneration kinetics. We did not observe significant changes in cytokine expression, likely due to the paucity of microglia representation in whole eye tissue from larva. We did see large fold-changes in expression for cytokines of interest, but as these did not reach levels of significance we declined to discuss them (the GEO submission provides all raw data). Finally, using qRT-PCR we evaluated changes in select inflammatory factors in response to D-Dex treatment and were able to resolve some differences (Supp. Fig.5, lines 420-423). Specifically, we saw upregulation of *Il6st* and *stat3*, downregulation of *tnfa* and no change in *mmp9*.

Minor points:

Comment 13: Figure 3 – it would be nice to have detailed histological data to go with the gross measures of fluorescence from NTR-YFP.

We have included histological data with respect to PCNA and BrdU labeling (see Fig. 4 and comment 8). These experiments were performed to add to our understanding of the mechanism by which D-Dex led to enhanced regeneration.

Comment 14: “super-accelerated regeneration” is over-stating the outcome.

We have changed terminology in the title to address this issue.

Reviewer #3 (Remarks to the Author):

This manuscript describes a study investigating the role that injury-induced inflammation plays in governing the regeneration of photoreceptors in zebrafish. Significantly, this manuscript also describes and validates the delivery of dendrimer-conjugated dexamethasone for microglia-specific, targeted immune suppression and a novel microscopy technique for imaging live zebrafish larvae. The technical innovations described in this manuscript are solid and will make an important contribution to the field. The underlying cause of the accelerated rod regeneration following DEX treatment is subject to multiple interpretations, which the authors should strive to include in revisions made to this manuscript.

Below are (mostly minor) points the authors should consider incorporating into any revisions of this manuscript.

Abstract

Comment 15: Line 48: As written, this sentence states that 'insights' 'stimulate limited regenerative responses.' This sentence should be rewritten for clarity.

We have rewritten this sentence as well as the following sentences that expand on it, this can be found at line 48-53

Introduction

Comment 16: Line 88: References should be added to support this sentence. This will aid readers, who may be grappling with this issue.

We have added a reference in support of this sentence which is now located at lines 99-100.

Results

Comment 17: Line 115: 'Post-ablation treatment suppresses microglia reactivity....' Actually, post-ablation Dex treatment alters the motility, and presumably also phagocytosis, of microglia. By virtue of the marked change from ramified to amoeboid morphology, it is clear that microglia in the Dex-treated retinas sense the death of the rod cells and are reactive. The authors may wish to be more circumspect regarding the conclusions (and interpretations) made from the cellular data.

We appreciate the suggestion to explain more carefully our interpretations of our imaging data. We now add additional measures of reactivity, all of which support D-Dex serving to repress microglia reactivity (see Comment 12 above). In the discussion, we highlight a key takeaway of the data: that microglia exhibit heterogeneous responses to rod cell death. This finding parallels other recent papers and we now devote a section of the discussion to the issue of microglia subtypes (lines 483-494).

Comment 18: Line 153: 'Notably,' I do not believe the authors provide sufficient evidence to state unequivocally that peripheral macrophages enter the eye when rod cells die. In the adult zebrafish retina, in contrast to mice, microglia reside among cells of the RPE. It is possible that some microglia migrating into the outer nuclear layer originate from above this layer, not outside the retina (is it even possible for macrophages to traverse Bruch's membrane?). The authors state in the legend to Supplemental Figure 1 that these microglia may migrate from the RPE. Microglia originating from the RPE layer is the most parsimonious interpretation of the images and videos and should be stated as such in the Results.

After reconsidering the data, we agree that more studies would be required to unequivocally demonstrate that this is occurring. We removed this interpretation of the data but do plan to investigate this question further in the future (see also Comment 6 above).

Comment 19: Line 156: Data is the plural of datum. Therefore, 'These data clarify....'

This statement has been removed.

Discussion

Comment 20: Following photoreceptor death in zebrafish, pro-inflammatory molecules originate from dying photoreceptors, Müller glia and microglia. Dendrimer-based targeting of dexamethasone to microglia may not impact the synthesis and release of pro-inflammatory molecules originating from cellular sources other than microglia, or even from microglia, themselves. The presence (and perhaps persistence) of pro-inflammatory molecules, even when Dex is present, may alter the kinetics of regeneration.

We thank the reviewer for this well-thought out suggestion. We have a growing interest in the signaling networks existing between dying retinal neurons, Müller glia and microglia and hope to explore this further in the future. In our revised manuscript, we have conducted bulk-RNA sequencing following treatment with D-Dex and this did tend to point us in alternative directions to looking at cytokine expression changes to explain the effects of D-Dex. However, as the tissue used in this case is the whole eye, it is very possible that we cannot resolve some of these changes in microglia specifically. In our reply to comment 11 from reviewer 2, we go into more depth explaining that we do have some evidence for D-Dex effecting the cytokines as well as the alternative paths provided by bulk-RNA seq. One of the initial reasons we believed that the cytokine effects in response to Dex would be prominent is a recent paper from the Hitchcock lab (Silva et. al 2020 published in *Glia*) did demonstrate using qRT-PCR that Dex induced cytokine changes in the adult zebrafish eye following a photoreceptor lesion (in this case Dex was given before injury rather than after). To address the deficit of significant findings from our bulk seq data, we provide data for qPCR quantification of some key inflammatory genes in Supp. Figure 5.

Additionally, in our reply in comment 9, we address the literature that suggests in other species the expression of the glucocorticoid receptor, *nr3c1*, is higher in the Müller glia rather than microglia and the effects of Dex on suppressing microglia reactivity may be a secondary impact following direct cell signaling effects in the Müller glia. We will further investigate this topic in the future.

Comment 21: Dex and D-Dex treatments after rod cell death suppress microglial migration

and phagocytosis. Phagocytosis, both by altering gene expression in macrophages and removing cellular debris, is a known mechanism for resolving inflammation. The absence of phagocytosis and/or persistence of photoreceptor corpses may alter the inflammatory environment within the outer retina in a manner that may accelerate the kinetics of rod regeneration.

We appreciate this comment on the likelihood of phagocytosis as another mechanism by which Dex and D-Dex may lead to altered microglia reactivity. We did evaluate phagocytosis in our prior report (White et al., PNAS, 2017). However, quantification is not trivial. To expand our analysis, we have included additional metrics of reactivity (see Comment 12).

Comment 22: Do the Dex treatments after rod cell death suppress inflammation in a manner similar to systemic pre-treatment with Dex, or do these treatments merely disable microglial migration? The authors conclude (line 248) that decreased migration speed is evidence of 'an immunosuppressive effect.' Are there data that support this statement? Are alternative interpretations possible?

In our prior publication where we investigated the effect of pre-Mtz Dex treatment on microglia reactivity and found that Mtz + Dex led to a decrease in the migration distance and displacement compared to Mtz alone (Figure 5b, White et. al 2017). This parallels our updated findings on post-Mtz Dex as noted in comment 10. With regards to sphericity, not previously measured, the improved spatiotemporal resolution provided by AO-LLSM uniquely enabled this measurement and thus while we expect to see similar effects in the pre-Mtz context, additional imaging experiments would need to be done to confirm this.

Comment 23: To be complete, the authors may wish to discuss the various sources of pro-inflammatory molecules in the zebrafish retina, the complexities underlying the inflammatory response during photoreceptor death, microglial phagocytosis and neuronal regeneration and the possibility that disabling migration of microglia may result in shifting the inflammatory milieu within the outer retina in a manner that is sufficient to accelerate rod regeneration.

We appreciate the suggestion to further discuss these topics. As noted in our response to comment 10 regarding investigating levels of inflammatory cytokines, we were unable to quantify significant differences with these types of factors using bulk RNA-seq data. As a result, our downstream analyses using CRISPR/Cas9 focused on non-inflammatory factors. To direct readers toward alternative papers that published evidence for shifts in pro-inflammatory factors and discussed them at length, as well as a review, we referenced these studies in the discussion (see lines 472-475).

Methods

Comment 24: Line 343: PTU is not without side effects. The authors may wish to determine whether or not PTU has any impact the innate immune system.

In all of our experiments, our methods for exposing the fish to PTU is uniform (exposure starting at 1 dpf) and thus we do not anticipate any changes between our conditions (Mtz only, no Mtz, etc) at 5 dpf and beyond when the studies in this paper begin. That being stated, we are aware of recent reports (for instance, Chen et. al 2021 in Autophagy) that suggests some of the effects that PTU can have. Our lab has had discussions previously about alternative methods to necessarily reduce the pigmentation of the eye such as injections of relatively low concentrations of Cas9 protein and sgRNAs targeting the *Tyrosinase* gene. These methods however are not without potential off-target concerns of their own. Thus, we will continue to monitor the literature and consider alternative options and/or conduct our own investigation into the effect of PTU on retinal cells.

Figures

Comment 25: In the photomicrographs of the outer nuclear layer, it appears that outer retina is on the left and inner retina is to the right, though this is only a guess. The orientation of the retina within the photomicrographs (and videos, if possible) should be made explicit

We have included these indications in the figure legend for figure 1, in each instance the ONL is to the left and the INL is to the right, see lines 213-214.

Reviewers' comments:

Reviewer #2 (Remarks to the Author):

The revisions to the manuscript are extensive and significant. The revisions satisfactorily address the comments of the reviewers.

Reviewer #4 (Remarks to the Author):

In this manuscript, the authors build on their previous work that indicates Dexamethasone (Dex)-mediated immunosuppression can alter rod regeneration kinetics in zebrafish larvae. These Dex-mediated effects depend on timing of treatment (pre- or post-ablation). In the previously published and current experiments, death of rods is induced in larvae expressing a transgene that results in NTR enzyme expression in rods, making them sensitive to the pro-drug mentronidazole (Mtz). Here, the authors attempt to target Dex to microglia through dendrimer (D) tagging to create D-Dex particles. They then use D-Dex in experimental manipulations after Mtz-mediated rod cell death and measure D-Dex effects on rod regeneration. They also provide some RNA-seq data and one figure of data probing two genes of interest that could be important in the effects of D-Dex. This review and comments pertain to the revised (second) version of the paper.

Major concerns:

The major conclusions, and impact, of the work presented is the dendrimer-based targeting of Dex to microglia, which is still not convincing. In other words, D-Dex did seem to speed up rod regeneration but it is not clear if D-Dex targeting microglia has a significant role in this effect. Conceptually, what is missing from the paper is any information of how dendrimers allow such targeting, either theoretically or with empirical evidence (for example, is a specific cellular receptor engaged?). Do the dendrimers facilitate uptake by phagocytic cells? Note that other glia are phagocytic (DOI: 10.1002/glia.24182, doi: 10.1096/fj.201801662R) and RPE cells are phagocytic, so if dendrimers facilitate general engulfment of D-Dex particles, then how can the authors conclude that the drug is indeed targeted only to microglia? If the authors cannot demonstrate this (see following comment below) then the claims of the effects of D-Dex on regeneration, and the title of the paper, should be significantly reworded.

With new data presented in this revision, prompted by previous reviewer comments, the authors attempt to support their claim that D-Dex is selectively targeted to microglia, based on imaging data provided in which D-Cy5 is injected. The authors follow Cy5+ signal in the retina, data presented in Figure 3. While D-Cy5 is indeed visible in putative phagocytic compartments in microglia, there is also D-Cy5+ puncta in other retinal locations outside of microglia. From the z-projection images of whole eyes, it is not clear where this signal is originating but it seems that it is visible in outer retina where RPE and photoreceptors themselves reside. This signal could even be within radial Muller glia. Thus, the authors cannot be certain that dendrimer tagging indeed exclusively targets microglia. Further, there is no basis to understand how visualization of Dex alone (if it were possible to tag) would differ from what they show in Fig 3. In addition, without injection of Cy5 alone, which could also look similar to what is seen in Fig 3, the authors cannot be certain that dendrimer tagging is the reason for accumulation of Cy5+ signal in microglia. Hence the authors cannot claim that dendrimers target localization of Dex to microglia (or other cell type) and that this particular cell targeting is the cause of the effects of D-Dex on rod regeneration.

In line with D-Dex targeting to microglia, previous review brought up the need to better show D-Dex effects on microglial reactivity. The authors now show analysis of microglial speed, displacement and sphericity; however these are inter-related parameters. Better support could be provided with other measurements such as phagocytic activity and changes to microglial gene expression. The authors provided bulk RNA-seq data, which they refer to in their response to a previous reviewer comment,

which they probed for changes in cytokine expression. However, this data still does not isolate effects on microglia as the dataset represents transcripts from all cell types in the preparation, and the bulk preparation could be a reason why they cannot parse out significant changes. The better approach would have been to FACS sort pure populations of microglia for RNA-seq/Differential gene expression between control and D-Dex treatments. The same is true for the genes examined by RT-qPCR.

For the genes targeted by CRISPR/Cas9, the authors should directly provide the gRNA sequences instead of directing readers to another paper. Further, the authors provide no data to support disruption of the genes that they targeted (*rnf2* or *kcjn13*). This is needed to interpret the effects of Cas9/sgRNA injection on retinal regeneration, especially if the authors are not going to follow up this screen with a mutant line. Further, and especially considering the comments above about D-Dex targeting to microglia, the authors should at least attempt to determine if these genes are expressed in microglia or another cell type in their different experimental conditions, as again the question of indirect vs direct effects of D-Dex, or which cell types are responding to D-Dex treatment, is the underlying component of this work.

Figure 2b data points need confidence intervals added. This is likely the reason that previous reviewers asked about sample sizes and statistics. Since the authors are presenting percentages from the total number of larvae analyzed for each data point, confidence intervals can be generated and added to the graph.

Other comments (minor):

Effects of Mtz on microglia was addressed in this revision with the authors examining microglia displacement, speed, and sphericity. Also important is microglial viability/numbers and phagocytic morphology. The authors may wish to know that this was previously examined in Blume et al. 2020 (Fig 1), showing that Mtz does not have effects on microglia that do not express NTR.
<https://doi.org/10.1002/dvdy.163>

In Fig 3e, the authors present co-localization data, however, it is difficult to understand what the numbers represent here. What is represented by the numbers on the y axis? This needs to be clarified for readers.

Author responses to the second set of reviews

We wish to thank Reviewer #2 for acknowledging that “the revisions to the manuscript are extensive and significant”, and that “the revisions satisfactorily address the comments of the [previous three] reviewers.”

We also wish to thank Reviewer #4 for the new comments. For reasons we detail below, and given the position of Reviewer #2 that we have satisfactorily addressed the comments of the previous three reviewers, we limited additional experimentation to one issue we felt could add significant value if successful (see comment 2). Below, we address each concern raised by reviewer #4, providing counter arguments where we disagreed and noting where we have revised the text to address the issues raised and/or to clarify aspects that led to confusion about our interpretation of the data.

Theme 1) Dendrimer targeting of reactive microglia

Reviewer Concern 1a: *“dendrimer-based targeting of Dex to microglia...is still not convincing”*

Response: We respectfully disagree. Firstly, our report is not novel in providing evidence of dendrimer targeting of reactive macrophage/microglia. That finding is supported by more than 60 published reports, including several focused specifically on dendrimers targeting microglia in the retina and brain (Iezzi et al., 2012; Kambhampati et al., 2015a and 2015b; Kambhampati et al., 2021; Cho et al., 2021; Pitha et al., 2023). For the sake of brevity, we previously referenced only two prior reports. We have now expanded the number of references to underscore the body of evidence in support of that finding. In particular, Pitha et al. provides numerous convincing histological examples of D-Cy5 dendrimers targeting retinal microglia in the context of a rat glaucoma model. What is novel in our report is that we assessed dendrimer targeting using intravital imaging, i.e., time lapse confocal microscopy *in vivo*, a technique larval zebrafish are particularly well-suited for. In Figure 3, we provide convincing evidence that dendrimers accumulate in highly motile retinal microglia. We feel the representative movie (Supp. Video 6) provides a particularly strong example that Cy5-labeled dendrimers (D-Cy5) not only localize within reactive microglia but track with them as they migrate throughout the retina.

Specific changes made: We have included the additional references listed above in support of dendrimers targeting reactive microglia and macrophages. We also clarify that accumulation in RPE cells has also been observed: “For example, we and others have shown that hydroxyl (poly)amidoamine (PAMAM) dendrimer nanoparticles enable targeting of drug conjugates to reactive microglia/macrophages and other phagocytic cells, such as retinal pigment epithelia (RPE) cells, in multiple inflammation-associated disease models²²⁻²⁸ (Lines 105-108). Additionally, we reference recent data demonstrating dendrimer-based targeting

of Dex to reactive retinal microglia in a rat model of glaucoma by Pitha et al., (Lines 113-114).

Lastly, the reviewer noted an incomplete understanding of the quantification of quantification used in figure 3e. We have added additional details to the manuscript to provide more clarity in the figure legend (Lines 325-328) and methods section (Lines 625-628). We hope that in light of the 60+ prior reports showing dendrimer uptake in reactive microglia and the data provided in Figure 3 and Supp. Video 6, that the reviewer can agree that dendrimers are indeed taken up by reactive microglia and that time lapse imaging provides an additional layer of nuance and rigor to that body of work.

Reviewer Concern 1b: *"the authors cannot be certain that dendrimer tagging indeed exclusively targets microglia", "there is also D-Cy5+ puncta in other retinal locations outside of microglia", lastly, ""Do the dendrimers facilitate uptake by phagocytic cells? Note that other glia are phagocytic (DOI: 10.1002/glia.24182, doi: 10.1096/fj.201801662R) and RPE cells are phagocytic, so if dendrimers facilitate general engulfment of D-Dex particles, then how can the authors conclude that the drug is indeed targeted only to microglia?"*

Response: We did not intend to suggest that dendrimers *"exclusively target microglia"* and indeed did not explicitly state that in the text. However, clearly, we need to do a better job of discussing dendrimer targeting specificity to avoid that perception. We have made several changes, detailed below, to address this issue. As the reviewer notes, our data shows that dendrimers accumulate in peripheral macrophages, consistent with our prior report showing they are also reactive to selective rod cell death (White et al., 2017), and also appear to label other unlabeled cells in the retina (potentially, retinal pigment epithelia, RPE, or Muller glia, MG). The z-projection data – i.e., where 3D volumes are presented as two-dimensional – includes tissue inside and outside the retina. This makes it difficult to determine relative localization, as the reviewer notes. In fact, we discussed this issue previously regarding peripheral macrophages that appear to enter the retina in our 2017 paper that are, in fact, macrophages within the cornea (White et al., 2017). In analyzing single z-plane images, we do indeed see evidence of dendrimer puncta that are not co-localized with microglia or macrophages. It is impossible to determine if these signals are intracellular or extracellular with the data presented. But as dendrimers have been shown to be taken up by RPE cells we have added a statement to the Results to bring attention to these signals and to conjecture about their localization to other cell types.

Specific changes made: We now note in the results section that we see evidence of D-Cy5 puncta that are not co-localized with microglia or macrophages. In the absence of labels for other cells types – e.g., RPE or MG (as the reviewer posits) – it is not possible to determine whether these signals are localized within or outside cells in these regions. Nevertheless, to ensure we have accounted for the possibility of dendrimer uptake in cells other than

microglia in these data we have added the following to the Results section, “Finally, we observed instances both within and outside of the retina where D-Cy5 puncta was not colocalized to transgenic microglia/macrophages. These signals suggest D-Cy5 association with other cell types such as dying rods, RPE, MG, or extracellular D-Cy5 that has yet to be taken up. We note that dendrimer uptake by RPE cells has been observed in prior reports²⁵, suggesting that this may be the source of this signal” (lines 309-313). With this change, and the statement added the Introduction that clarifies this point, we hope the reviewer will find adequate changes have been made regarding the possibility of dendrimers targeting cells other than microglia in our data.

Reviewer Concern 1c: *“...without injection of Cy5 alone, which could also look similar to what is seen in Fig 3, the authors cannot be certain that dendrimer tagging is the reason for accumulation of Cy5+ signal in microglia”...“Further, there is no basis to understand how visualization of Dex alone (if it were possible to tag) would differ from what they show in Fig 3.”*

Response: We thank the reviewer for these comments but believe we included the more relevant control in the context of our study which was testing and quantifying D-Cy5 localization in the unperturbed retina where no rod cell ablation was induced (Fig. 3d,e). To address the reviewers proposed control of injecting Cy5 alone, we found a report where the localization of Cy5.5 injected into larval zebrafish had been investigated (Liang et al., 2018, “Enhanced blood–brain barrier penetration and glioma therapy mediated by T7 peptide-modified low-density lipoprotein particles”). Figure 4 of their study demonstrates that free Cy5.5 was not able to cross the blood-brain barrier being found only in the vasculature of the larval fish. Given that entry into the retina would require crossing the blood-retinal barrier, we feel this data adequately addresses the issue of whether Cy5 alone would be capable of localizing to CNS microglia either in the brain or retina.

In an attempt to add additional data regarding the specificity of D-Dex in targeting reactive microglia compared to Dex alone. We took the reviewers suggestion in attempting to label Dexamethasone in the retina. Using an anti-Dex antibody published in Perez-Orsola et al. in 2021 where the authors show quite robust labeling of Dex following injection into mice (see image below).

Unfortunately, in our hands, after performing multiple assays to label injected Dex in our transgenic fish following the experimental paradigm in figure 3 (+/-Mtz larvae sacrificed at 4 hours post pericardial injection of Dex or D-Dex), we unfortunately were not able to get robust labeling using this resource. Very rarely we did observe co-localization of the anti-Dex antibody with microglia in D-Dex injected fish (see figure below; green asterisks indicate examples of cyan (Dex) present in red cells (microglia) while orange asterisks indicate Dex only labeling). However, this data was too sparse to allow quantified comparisons. Accordingly, we ask that additional experimentation regarding this concern be considered beyond the scope of this report due to the absence of adequate resources.

Theme 2) Necessity of microglia targeting for D-Dex effects on regeneration:

Reviewer Concern 2: *"...D-Dex did seem to speed up rod regeneration but it is not clear if D-Dex targeting microglia has a significant role in this effect" ... "it seems that it [dendrimer puncta] is visible in outer retina where RPE and photoreceptors themselves reside. This signal could even be within radial Muller glia." ... "Hence the authors cannot claim that dendrimers target localization of Dex to microglia (or other cell type) and that this particular cell targeting is the cause of the effects of D-Dex on rod regeneration." ... "If the authors cannot demonstrate this (see following comment below) then the claims of the effects of D-Dex on regeneration, and the title of the paper, should be significantly reworded."*

Response: Given that: 1) RPE cells have not been shown to have a role in retinal regeneration in zebrafish, 2) we nor any other group studying dendrimer localization in retinal disease models has reported dendrimer uptake in MG cells, 3) evidence provided here that post-injury dexamethasone (Dex) has effects on microglia reactivity consistent with accelerated resolution to a homeostatic state, and 4) the body of published evidence supporting microglia as regulators of retinal regeneration in zebrafish outlined above, we feel that our conclusion that dendrimer-enabled microglia targeting of Dex is linked to the enhanced regenerative effects observed is reasonable, warranted, and well within the bounds of a reasonable interpretation of the data presented.

To put this data into further context, our interpretation that post-ablation D-Dex accelerated regeneration is mediated by targeting of Dex to microglia is based on new data showing effects of post-ablation Dex on microglia reactivity (Fig. 1 and Supp. Videos 1-4) and previous studies from our group and others that have strongly implicated microglia in regulating retinal regeneration in zebrafish. Our prior work implicated microglia in regulating Müller glia (MG) activation and the kinetics of the regenerative process in the zebrafish retina (White et al., PNAS, 2017). Subsequent studies have been consistent with that finding (Mitchell et al., J Neuroinflammation, 2018; Mitchell et al., Sci Rep, 2019; Silva et al., Glia, 2020; Leach et al., PNAS, 2021; Iribarne and Hyde, Front Cell Dev Biol, 2022), leading many in the field to conclude that microglia regulate MG activation and the retinal regenerative process more broadly. In particular, we showed that co-ablation of microglia and rod photoreceptors delayed both MG and progenitor cell proliferation and rod cell regeneration kinetics. Moreover, application of dexamethasone (Dex) prior to rod cell ablation abolished microglia reactivity, robustly inhibited MG and progenitor cell proliferation, and phenocopied the delayed rod cell regeneration kinetics effect of microglia co-ablation. Conversely, the effects of exposing fish to Dex 24 hrs after induction of rod cell loss led to a surprising increase in the kinetics of rod cell regeneration (White et al., 2017). We attributed this result to accelerated resolution of microglia reactivity, positing a biphasic role for microglia during regeneration: initial reactivity being required to stimulate MG

activation but subsequent reactivity negatively influencing the pace of retinal cell regeneration.

Here, we now show new data consistent with that hypothesis, namely that post-ablation Dex treatment reduces established measures of microglia reactivity – i.e., reduces migration speed and translocation to non-reactive control levels. The reviewer states that these are inter-related measures and asks for additional metrics (i.e., rate of phagocytosis or changes in gene expression). The former is difficult to measure once phagocytosis is already ongoing – as is this case for the post-ablation treatment paradigm – and the latter is rendered difficult in larval zebrafish due to the low number of microglia present at this stage. Moreover, by directly measuring effects on microglia behavior *in vivo* using intravital time lapse imaging, our study provides a more nuanced perspective than more common static end point analyses, allowing us to resolve potential effects on the resolution kinetics of microglia reactivity. Accordingly, and in light of having satisfied reviewer #2 with respect to the requests of the prior three reviewers, we ask that our report go forward for publication as titled and that we be allowed the intellectual freedom to interpret our data in a manner consistent with prior reports and the data presented.

Changes made: In order to ensure readers are given a clear explanation of how we interpret this study, and how it relates to the other related work in the field, we have now added to our discussion a succinct version of our above comment, we now state- “Here, combining *in vivo* time-lapse imaging and immunohistology to investigate cellular responses, we found evidence that post-injury Dex inhibited microglia reactivity and/or accelerated microglia resolution, and increased proliferation of presumptive MG and MG-derived progenitors. Conjugation to dendrimer nanoparticles served to target Dex to reactive microglia (and possible other phagocytes such as RPE), reduced systemic toxicity and improved the pro-regenerative effects of post-injury immunosuppression. Combined with recent findings that: 1) macrophages/microglia have been implicated in multiple regenerative paradigms in the zebrafish retina^{45,48,49,51,52}, 2) selective rod cell ablation does not lead to invasion of peripheral macrophages¹⁶ and, 3) that glucocorticoids, such as Dex, are established immunosuppressants⁵³, we interpret the enhanced regenerative effects of dendrimer-Dex conjugates to likely be a result of enhanced resolution of microglia reactivity. Still, given that MG in species other than fish have been shown to express the glucocorticoid receptor⁵⁴ and that glucocorticoids have been shown to have effects on retinal stem cells^{54,55}, we cannot rule out an additional role for effects of Dex or dendrimer-Dex on MG cells directly” (Lines 478-491).

We thank the reviewer for the push to more directly explain why we are interpreting our data in the manner that we do, while also giving attention to other possibilities (such as additional effects Dex may have on MG).

Theme 3) Mechanism of dendrimer targeting of microglia

Reviewer Concern 3: *"Conceptually, what is missing from the paper is any information of how dendrimers allow such targeting, either theoretically or with empirical evidence (for example, is a specific cellular receptor engaged?)."*

Response: We appreciate the reviewer's interest in a mechanistic understanding of how dendrimers target phagocytes in the absence of a targeting ligand. In fact, we are interested in the same question and plan to use a large-scale genetic screen to identify specific factors that mediate this phenomenon. However, given: 1) defining the molecular mechanism of dendrimer targeting is an entire study onto itself that will be no small feat, 2) ample evidence of dendrimer targeting of microglia in several model systems (see 60+ published reports discussed above), and 3) the compelling reduction in Dex-associated toxicity (consistent with prior studies) and enhanced regenerative effects we observe with D-Dex (our novel finding), we ask that we be allowed to pursue this question as an independent study. The novelty of testing dendrimers in the zebrafish retina and seeing significant effects on rod photoreceptor regeneration kinetics, we feel, is compelling enough to warrant publication at this point in our investigations.

Theme 4) Minor points, changes, discussion topics

:

Reviewer Concern 4a: Confidence intervals requested for figure 2b:

Changes made: We have now generated and included confidence intervals to the plot in figure 2b.

Reviewer Concern 4b: Reference to prior study where Mtz alone was measured for effect on phagocytic activity in larval fish (Blume et al):

Changes made: We thank the reviewer for pointing out this study where Mtz alone was shown to have no effect on phagocytosis, we have added this as a sentence and reference where we share our own results on morphological measures in Supp. Fig. 3 (Lines 234-236).

Reviewer Concern 4c: Provide gRNA sequences for those targeted by CRISPR and provide more reasoning as to why those genes were selected. Discuss known expression patterns of those genes if possible:

Response: Sequences for gRNA sequences are now provided. We evaluated the effects of knocking down the expression 4 genes implicated in our bulk RNA-seq data comparing the Mtz +D-Dex to Mtz alone. To identify the top candidates, we carried out a literature search of the DEGs showing the most statistically significant changes. *Kcnj13* and *rnf2*, the two

genes we were able to test for effects on regeneration as they did not impair larval development, were selected as both were shown to be expressed in the eye and the immune system based on data available in the ZFIN database. As we note in supplemental figure 5, we saw some changes in known immune regulators such as *il6st*, *stat3* and *tnfa*, but since these genes had a much lower fold change that was calculated as not statistically significant we felt it more prudent to test factors showing statistically significant changes in expression.

Specific changes made: We now provide all gDNA sequences in Supp. Table 2. In the results section, we previously noted that the tested genes were chosen because they were identified DEGs in the bulk RNA data- we have now added additional clarification that these were the top upregulated DEGs when comparing Mtz +D-Dex to Mtz alone as well as the average fold change of these genes. Finally, we note that ZFIN supports the expression of *kcnj13* and *rnf2* in the eye of zebrafish as well as playing a role in the immune system (Lines 445-450). This information is also now in the methods section under CRISPR/Cas9 redundant gene targeting. (Lines 781-787).

Reviewer Concern 4d: *“Gene expression data does not isolate effects on microglia due it being bulk in nature, FACS sorting would be a better approach in this case”*

Response: We agree that alternative methods may have done a better job of isolating microglia specific effects, such as FACS sorting prior to bulk RNA-sequencing or single cell transcriptomics. We have indeed performed scRNA-seq and ATAC-seq assays in other retinal paradigms and note that resolving gene expression changes in microglia is complicated by the low number of these cells in the larval retina. In addition, the process of cell dissociation necessary for FACS has now been shown to activate macrophages/microglia in and of itself (Marsh et al. 2022 Nature Neuroscience). This data makes clear that “reactive” macrophage/microglia gene changes specific to the paradigm under study may be difficult to resolve from technical background caused by dissociation alone. Finally, these techniques have an additional downside, in that we then would be missing other cell types important to the regenerative process such as Muller glia, dying rods, etc. Our lab has a great interest in performing additional sequencing experiments to isolate gene expression changes regulating the immune response to damage and is preparing additional manuscripts on this topic. With respect to the current manuscript, and given the technical issues outlined above, we feel that the data presented are sufficient for the tests performed and conclusions reached.